# DexFlyWheel: A Scalable and Self-improving Data Generation Framework for Dexterous Manipulation

**Kefei Zhu**[1,2,4,*], **Fengshuo Bai**[4], **YuanHao Xiang**[4], **Yishuai Cai**[4], **Xinglin Chen**[4], **Ruochong Li**[4],
**Xingtao Wang**[1,5], **Hao Dong**[3], **Yaodong Yang**[3,4,†], **Xiaopeng Fan**[1,5,†], **Yuanpei Chen**[2,3,4,†]

[1]Harbin Institute of Technology  [2]PsiBot  [3]Peking University
[4]PKU-Psibot Lab  [5]Harbin Institute of Technology Suzhou Research Institute

kefei.zhu321@gmail.com  yaodong.yang@pku.edu.cn  fxp@hit.edu.cn  yuanpei.chen312@gmail.com
Project Page: https://DexFlyWheel.github.io

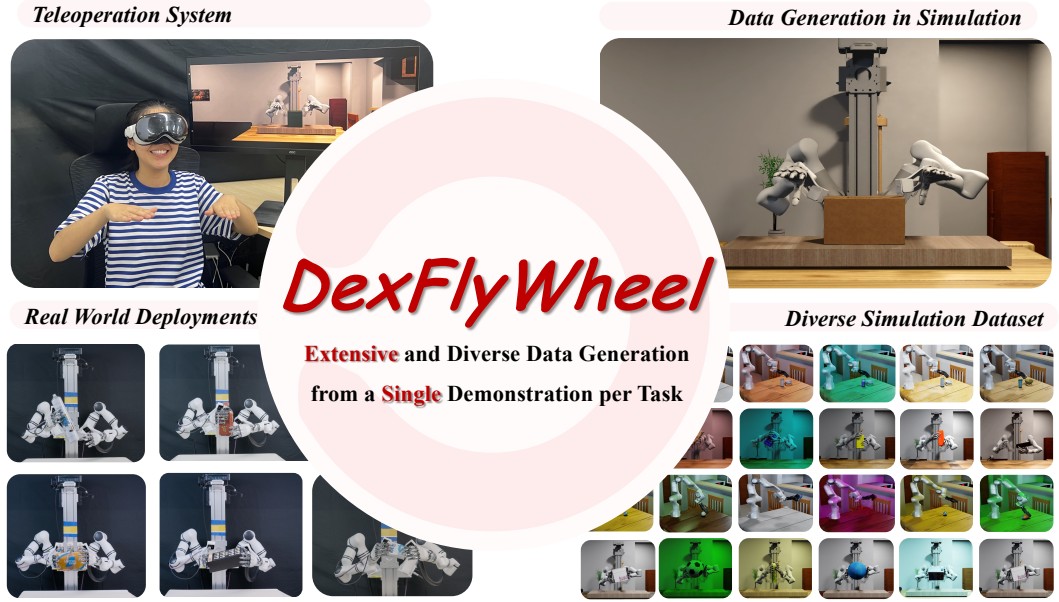

Figure 1: **Scaling dexterous manipulation data** — DexFlyWheel generates diverse, high-quality dexterous manipulation data for challenging tasks. Our generated dataset enables policies to generalize to unseen scenarios and successfully transfer to the real world.

## Abstract

Dexterous manipulation is critical for advancing robot capabilities in real-world applications, yet diverse and high-quality datasets remain scarce. Existing data collection methods either rely on human teleoperation or require significant human engineering, or generate data with limited diversity, which restricts their scalability and generalization. In this paper, we introduce DexFlyWheel, a scalable data generation framework that employs a self-improving cycle to continuously enrich data diversity. Starting from efficient seed demonstrations warmup, DexFlyWheel expands the dataset through iterative cycles. Each cycle follows a closed-loop pipeline that integrates Imitation Learning (IL), residual Reinforcement Learning (RL), rollout trajectory collection, and data augmentation. Specifically, IL extracts human-like behaviors from demonstrations, and residual RL enhances

---

[†]Corresponding author
[*]This work was completed during an internship at PsiBot.

39th Conference on Neural Information Processing Systems (NeurIPS 2025).

policy generalization. The learned policy is then used to generate trajectories in simulation, which are further augmented across diverse environments and spatial configurations before being fed back into the next cycle. Over successive iterations, a self-improving data flywheel effect emerges, producing datasets that cover diverse scenarios and thereby scaling policy performance. Experimental results demonstrate that DexFlyWheel generates over 2,000 diverse demonstrations across four challenging tasks. Policies trained on our dataset achieve an average success rate of 81.9% on the challenge test sets and successfully transfer to the real world through digital twin, achieving a 78.3% success rate on dual-arm lift tasks.

# 1 Introduction

Learning from Demonstration (LfD) [1] has become increasingly prevalent in robotics. Recent works has shown that training large models with extensive datasets can achieve more challenging tasks and better generalization [2–8]. In dexterous manipulation particularly [9–12], the higher degrees of freedom and richer contact interactions demand larger, more diverse and higher-quality datasets. However, collecting such datasets remains a considerable bottleneck. Human teleoperation approaches require significant human effort and typically constrain data collection to laboratory settings, which limits the scalability of data collection. While portable motion capture devices [13] can collect data in-the-wild, they still require substantial human involvement and suffer from cross-embodiment gap. Recently, simulation has emerged as a promising solution to address data challenges in robotics [14–19]. It offers numerous advantages: parallel data collection at scale, easy modification of robot embodiments and sensor configurations, and domain randomization for data augmentation. However, existing simulation-based approaches, such as optimization or heuristic planning methods [20], LLM-driven methods [19, 21–23], and purely RL-based methods [18, 24–26], struggle with the high-dimensional complexity of dexterous manipulation and often produce low-quality trajectories.

Given these simulation challenges, researchers have begun exploring the teleoperation with replay-mechanism [14, 27], where humans teleoperate simulated robots to collect training data and then use spatial transformations to synthesize new trajectories. Although this approach captures relatively high-quality data with simulation-based augmentation, it has several fundamental critical limitations: *(1) Inability to Explore Novel Manipulation Strategies.* By replaying human demonstrations, these methods confine exploration to existing behaviors, restricting data to the scope of the original demonstrations and hindering generalization to novel scenarios. *(2) Insufficient Data Diversity.* Since these methods primarily apply spatial augmentations, the generated datasets often exhibit insufficient variability in object geometries and environments, inherently constraining the generalization of learned policies. These limitations motivate us to rethink the role of human demonstrations in data generation pipelines. We observe that manipulating different objects typically induces only minor changes in the manipulation trajectories. This suggests regarding human demonstrations not merely as replay data, but as strong behavioral priors that can guide exploration in novel scenarios.

Building on this insight, we propose **DexFlyWheel**, a scalable and self-improving data generation paradigm for dexterous manipulation. Our framework features two key design: **IL + residual RL for Human-like and Diverse Data Generation.** DexFlyWheel combines IL to learn human-like behaviors from demonstrations with residual RL to adapt these priors to novel scenarios, particularly when manipulating different objects, thereby generating diverse and human-like data. **A Dexterous Manipulation Data Flywheel.** Inspired by iterative self-improvement in LLMs [28, 29], we design a data flywheel for dexterous manipulation. Specifically, IL and residual RL are combined with policy rollouts and data augmentation to form a self-improving cycle. At each iteration, the policy generates trajectories, which are then augmented in progressively more diverse scenarios and subsequently fed into the next iteration. This cycle produces a *flywheel* effect, progressively expanding data diversity, enhancing policy generalization, and evolving into a robust, generalizable data generation agent.

**In summary**, our main contributions include:

- We propose **DexFlyWheel**, a scalable and self-improving data generation framework for dexterous manipulation. By combining IL with residual RL and leveraging data augmentation within a self-improving flywheel mechanism, our framework efficiently produces diverse, high-quality demonstrations while preserving human-like behavior patterns. This alleviates the scarcity of dexterous manipulation data and provides a solid foundation for training generalizable policies.

- We demonstrate the effectiveness of our framework on four dexterous manipulation tasks. Starting from a single human demonstration per task, DexFlyWheel generates over 2,000 successful demonstrations across 500+ diverse scenarios. The *flywheel* effect of our framework progressively expands data diversity, enabling it to significantly outperform baseline data generation methods.

- We validate that policies trained on our generated data achieve an average success rate of 81.9% on challenging test sets, significantly outperforming policies trained with baselines. Furthermore, our policies transfer to a real-world dual-arm robot system via digital twin, achieving a 78.3% success rate on the dual-arm lift task and a 63.3% success rate on the dual-arm handover task.

## 2 Related Work

**Dexterous Manipulation.** Dexterous manipulation with multi-fingered robotic hands remains a significant challenge in robotics [30–35], largely constrained by high-quality demonstration data scarcity. While the prevailing approach employs reinforcement learning to develop manipulation skills, this method frequently encounters efficiency limitations and exploration challenges [36–39]. Researchers have explored human video demonstrations [40–47], but morphological differences between human and robotic hands create substantial transfer barriers. Human teleoperation has emerged as a promising alternative for collecting expert trajectories for imitation learning [14, 27, 48–51], effectively capturing expert actions in native robot morphology. Nevertheless, existing approaches still struggle with data collection efficiency or require extensive human engineering, emphasizing the need for high-quality dexterous manipulation datasets.

**Robotic Data Generation in Simulation.** Current approaches for collecting robotic demonstrations in simulation face significant limitations when applied to dexterous manipulation. Motion planning-based methods, while effective for gripper-based systems [15, 20, 24, 52–54], struggle with the high-dimensional action space and complex contact dynamics of multi-fingered manipulation. LLMs-driven methods [19, 21–23] can generate high-level command, but they demonstrate limitations when confronted with high-degree-of-freedom dexterous hands, unable to provide the fine-grained guidance necessary for coordinated finger-level control. Other pipelines are designed specifically for grasping [52], and thus do not generalize well to more complex dexterous tasks. RL-based methods have also been widely adopted [36, 24–26, 18, 53, 55, 56], yet purely RL-trained policies often exhibit non-human-like behaviors, leading to less robust manipulation and increased sim-to-real transfer challenges. Moreover, RL faces exploration difficulties and relies heavily on reward engineering, which is particularly acute in dexterous manipulation. Replay-based methods [14, 27, 57] attempt to edit existing demonstrations to new scenarios but face fundamental scalability constraints, as they merely implement spatial transformations of recorded trajectories without the ability to explore novel manipulation strategies beyond the original demonstrations. For example, when an object's geometry changes significantly—e.g., from a sphere to a cuboid—replay-based methods struggle to adapt finger trajectories. These collective limitations underscore the critical need for more efficient and scalable methods to generate diverse, high-quality dexterous manipulation data in simulation.

## 3 Task Formulation

To address the challenge of generating high-quality synthetic data for robotic manipulation tasks, we train policy models for each manipulation task in simulation environments and use these policies to collect demostrations. We formulate each manipulation task as a Markov Decision Process (MDP) $\mathcal{M} = (\boldsymbol{S}, \boldsymbol{A}, \pi, \mathcal{T}, R, \gamma, \rho, G)$, where $\boldsymbol{S}$ is the state space, $\boldsymbol{A}$ is the action space, $\pi$ is the agent's policy, $\mathcal{T}(\boldsymbol{s}_{t+1}|\boldsymbol{s}_t, \boldsymbol{a}_t)$ is the transition distribution, $R$ is the reward function, $\gamma$ is the discount factor, and $\rho$ is the initial state distribution. The policy $\pi$ conditions on the current state $\boldsymbol{s}_t$, and generates robot action distributions $\boldsymbol{a}_t$ to maximize the likelihood between the future object states $(\boldsymbol{s}_{t+1}, \boldsymbol{s}_{t+2}, \ldots, \boldsymbol{s}_{t+T})$.

## 4 Method

In this section, we begin with an overview of the DexFlyWheel architecture (Section 4.1) and then detail the two-stage data generation pipeline (Sections 4.2 and 4.3). Together, these components enable scalable collection of diverse and high-quality dexterous manipulation demonstrations.

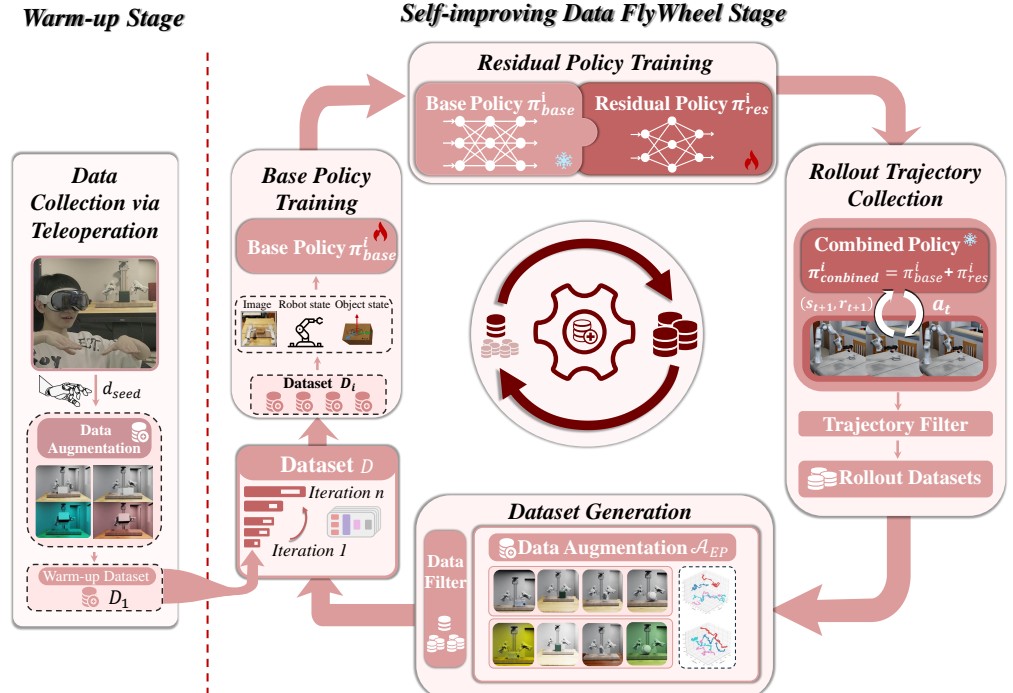

Figure 2: **DexFlyWheel Framework Overview.** The framework has two stages: **a warm-up stage *(left)*** and **a self-improving data flywheel stage *(right)***. In the warm-up stage, seed demonstrations from VR teleoperation are augmented to form the initial dataset $\mathcal{D}_1$. The data flywheel stage operates as a closed-loop cycle with four key components:(1) base policy $\pi_{base}$ training to capture human-like behaviors, (2) residual policy $\pi_{res}$ training to enhance generalization, (3) combined policy $\pi_{combined}$ rollouts to generate new trajectories, and (4) data augmentation to further diversify the dataset. As the flywheel iterates, both data diversity and policy capability continuously improve.

## 4.1 Overview

DexFlyWheel aims to generate diverse and high-quality data across various scenario configurations, providing broad coverage of objects, environments, and spatial variations, while only starting with minimal human demonstrations. Figure 2 illustrates the overall architecture of our framework.

**Warm-up stage.** A single human demonstration $d_{\text{seed}}$ is collected via a VR-based teleoperation system. This seed demonstration is then processed using a multi-dimensional data augmentation module $\mathcal{A}_{\text{EP}}$. Given the human demonstration $d_{\text{seed}}$, the augmentor generates new demonstrations with diverse environment and spatial variations to produce the initial dataset $\mathcal{D}_1$.

**Self-improving Data FlyWheel stage.** We design a data flywheel mechanism to progressively enhance both data diversity and policy performance. This stage comprises multiple iterations $i = \{1, 2, \ldots, n-1\}$, at each iteration $i$, the following steps are executed: *(1)* An imitation learning policy $\pi_{\text{base}}^i$ is trained on dataset $\mathcal{D}_i$ (with $\mathcal{D}_1$ used when $i = 1$). *(2)* To improve generalization to novel objects, a residual reinforcement learning policy $\pi_{\text{res}}^i$ is trained on top of the frozen $\pi_{\text{base}}^i$, yielding a combined policy $\pi_{\text{combined}}^i = \pi_{\text{base}}^i + \pi_{\text{res}}^i$. *(3)* The combined policy is deployed in simulation to generate demonstrations under various object configurations, forming a high-quality rollout dataset $\mathcal{D}_{\text{O}}^i$. *(4)* Finally, $\mathcal{D}_{\text{O}}^i$ is further augmented by $\mathcal{A}_{\text{EP}}$ with environment and spatial variations to produce the dataset $\mathcal{D}_{i+1}$ used in the next iteration.

## 4.2 Warm-up Stage

The first stage aims to generate an initial dataset $\mathcal{D}_1$ via data augmentation module $\mathcal{A}_{\text{EP}}$, starting from a single human demonstration $d_{\text{seed}}$. This warm-up stage including two operations:

**Data Collection via Teleoperation.** To bootstrap the framework with high-quality seed data, we design a VR-based teleoperation system implemented in simulation using Apple Vision Pro [50] to accurately track human hand, wrist, and head poses. Since large-scale data collection requires heavily

human effort, we only need a single demonstration, denoted as $d_{\text{seed}}$. This demonstration serves as the sole seed for subsequent data generation. This makes our method highly efficient in terms of human resources while maintaining the quality and diversity of the generated data.

**Data Augmentation.** To scale and diversify our dataset, we introduce data augmentation module $\mathcal{A}_{\text{EP}}$, which builds upon the MimicGen framework [27] and extends it to support multi-dimensional data augmentation across various environments and spatial configurations. It can efficiently augment source dataset $\mathcal{D}$ through trajectory editing and simulation domain randomization. $\mathcal{A}_{\text{EP}}$ takes $\mathcal{D}$ and augmented scenario configurations $\mathcal{C}_{\text{aug}}$ as input, and outputs the augmented dataset $\mathcal{D}'$. In the warmup phase, this process is applied to the seed demonstration: $\mathcal{D}_1 = \mathcal{A}_{\text{EP}}(d_{\text{seed}}; \mathcal{C}_1)$. Through this warmup phase, we establish a foundation of diverse manipulation datasets that serves as the starting point for our iterative data flywheel mechanism.

### 4.3 Self-improving Data FlyWheel Stage

The second stage implements a closed-loop data flywheel mechanism that iteratively expands data diversity across objects, environments and spatial generalization dimensions. At each iteration $i \in \{1, 2, ..., n-1\}$, the data flywheel performs four key operations:

**Base Policy Training.** Given the dataset $\mathcal{D}_i$ from the previous iteration, we employ diffusion-based policy [58] as the base policy $\pi_{\text{base}}$ to learning dexterous manipulation skill, obtaining a strong base policy for subsequent modules. At each step $t$, the base policy $\pi_{\text{base}}^i$ takes the state $s_t = \{s_t^{\text{vis}}, s_t^{\text{obj}}, s_t^{\text{prop}}\}$ as inputs, where $s_t^{\text{vis}}$ represents visual input from camera, $s_t^{\text{obj}}$ contains object state information including 6D pose (position and orientation) and velocities, and $s_t^{\text{prop}}$ includes robot proprioception data consisting of joint positions, velocities, and end-effector poses. The policy outputs a sequence of robots actions $(a_t, a_{t+1}, \ldots, a_{t+H})$, where $H$ represents the prediction horizon, and each action $a_t$ consists of the end-effector 6D pose and target joint angles. Implementation details of the policy parameters are provided in Appendix A.1.

**Residual Policy Training.** Generalizing to novel objects remains a key challenge in imitation learning for robotic manipulation, which often suffers from limited data. We observe that manipulating different objects induces only small changes in the manipulation trajectories[1], suggesting that a well-initialized policy can only require fine-grained adjustments to adapt to new objects. Based on this observation, we propose a residual reinforcement learning framework that builds upon the base policy. Specifically, we train a residual policy $\pi_{\text{res}}^i$ that takes object state $s_t^{\text{obj}}$ and robot proprioception $s_t^{\text{prop}}$ as inputs and generates correction actions $\triangle a = (\triangle a_t, \ldots, \triangle a_{t+H})$. These corrections, scaled by $\alpha$, are added to the base policy actions to form the combined policy $\pi_{\text{combined}}^i = \pi_{\text{base}}^i + \alpha \cdot \pi_{\text{res}}^i$, where $\tilde{a}_t = a_t + \alpha \cdot \triangle a_t$ at each timestep. This approach allows the residual policy start from a reasonable robot actions from $\pi_{\text{base}}^i$ and focus on learning the fine-grained refinements to generalize objects. Implementation details are in Appendix A.2.

To stabilize exploration, we employ the progressive schedule from [59], defining the combined policy during training as:

$$\pi_{\text{combined}}(s) = \begin{cases} \pi_{\text{base}}(s) + \alpha \cdot \pi_{\text{res}}(s) & \text{with probability } \epsilon \\ \pi_{\text{base}}(s) & \text{with probability } 1 - \epsilon \end{cases} \tag{1}$$

where $\epsilon$ serves as a mixing coefficient that linearly increases from 0 to 1 over $T$ steps, gradually shifting control from the base to the residual policy.

**Rollout Trajectory Collection.** In this module, we employ the frozen combined policy $\pi_{\text{combined}}^i = \pi_{\text{base}}^i + \alpha \cdot \pi_{\text{res}}^i$ to perform rollouts in simulation under randomized object configurations: $\mathcal{D}_{\text{O}}^i = \{d_j = \{(s_t, a_t)\}_{t=0}^{T-1} | d_j \sim \pi_{\text{combined}}^i\}_{j=1}^K$, where we collect $K$ high-quality trajectories by filtering based on task success. This rollout strategy achieves robust object generalization, while geometry-unaware trajectory editing methods fail to adapt [14].

**Data Augmentation.** In this module, we employ the previously introduced data augmentation module $\mathcal{A}_{\text{EP}}$ to efficiently augment data in various environment and spatial configurations. Taking the dataset $\mathcal{D}_{\text{O}}^i$ and augmented scenario configurations $\mathcal{C}_{\text{aug}}^{i+1}$ as input, the module produces an expanded dataset: $\mathcal{D}_{i+1} = \mathcal{A}_{\text{EP}}(\mathcal{D}_{\text{O}}^i; \mathcal{C}_{\text{aug}}^{i+1})$. The expanded dataset $\mathcal{D}_{i+1}$ is used to train an improved base policy, which serves as the foundation for the next iteration.

---

[1]See Appendix A.7 for details.

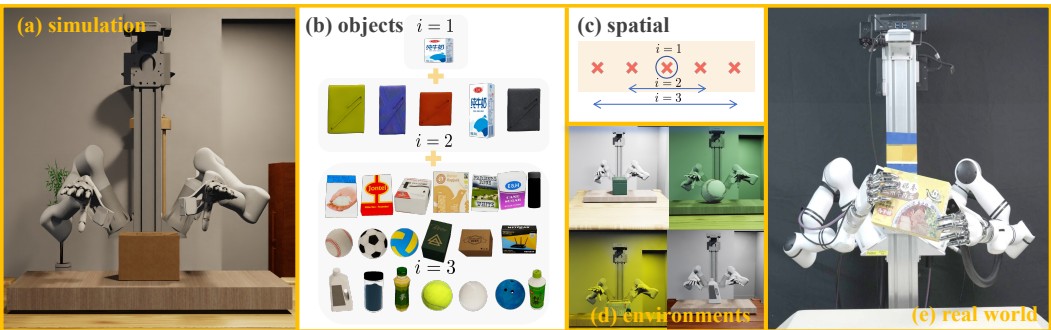

Figure 3: **Experiment Setup.** Taking the dual-arm robot system as an example, **(a)** Our simulation environment. **(b)** Object diversity expansion across iterations, progressing from a single object (i=1) to geometrically similar objects (i=2) and diverse geometries and physical properties objects (i=3). **(c)** Spatial diversity, showing the spatial arrangements. **(d)** Environment diversity, including variations in lighting conditions and tabletop appearances. **(e)** Real-world environment.

## 5 Experiments

The experiments are designed to answer the following research questions:

**Q1: Data Flywheel Effect.** *How does DexFlyWheel exhibit a self-improving data flywheel in dexterous manipulation, continuously enhancing data diversity and policy generalization? (Section 5.2)*

**Q2: Policy Performance and Data Generation Efficiency.** *How does DexFlyWheel compare with baselines in policy performance, data generation robustness, and time efficiency (Section 5.3)?*

**Q3: Component Contribution.** *How does each component of DexFlyWheel contribute quantitatively to the overall system performance? (Section 5.4)*

**Q4: Real-World Deployment.** *How does DexFlyWheel enable the real-world deployment of bimanual dexterous robot systems? (Section 5.5)*

### 5.1 Experimental Setup

**Tasks and Robots.** We evaluate our framework across four dexterous manipulation tasks on both single-arm and dual-arm robot settings: (1) **Grasp (single-arm):** The robot must grasp the target object and lift it to a height greater than 0.2 meters from the tabletop. (2) **Pour (single-arm):** The robot manipulates a source container to transfer its contents into a target container, requiring controlled pouring of the contained objects from one receptacle to another. (3) **Lift (dual-arm):** The robot performs collaborative manipulation using its two arms to synchronously lift an object to a minimum height of 15 cm. (4) **Handover (dual-arm):** The robot performs an intra-agent handover by transferring an object from one hand to the other through a stable, coordinated motion.

For the single-arm Grasp and Pour tasks, we use a Franka Emika Panda robot arm equipped with an Inspire robotic hand. For the dual-arm Lift and Handover tasks, we use a 7-DoF Real-Man RM75-6F arm paired with a 6-DoF PsiBot G0-R hand. Dual-arm robot is equipped with a RealSense D435 camera mounted on its head, which provides a first-person perspective.

**Data Collection and Environment.** For each manipulation task, we collected only a single demonstration trajectory using VR teleoperation as our minimal seed data. To ensure the generation of high-quality data, we employed OmniGibson [60] as our simulation platform, leveraging its realistic rendering to generates high-quality data. We prepared 80 distinct objects across various categories and 12 different environments with varying lighting conditions, tabletop appearances. we set the number of iterations to $i = \{1, 2, 3\}$. For each task, we generated 20, 100, and 500 trajectories in the three iterations, respectively. Simuation setup is visualized in Figure 3. More detailed data collection and environment setup are provided in Appendix A.3

**Evaluation Design.** We evaluate our method using two criteria: (1) **data diversity**: the number of object variations $O$, environment variations $E$ and spatial variations $P$ in our generated dataset

$D_i$ in each iteration, and the total number of scenarios ($O \times E \times P$) that our pipeline can cover; (2) **generalization performance**: the Success Rate ($SR$) of task execution when policies trained on our generated datasets $D_i$. It is calculated as the ratio of successful task completion to the total attempts. Specifically, we construct two types of test sets. First, the multi-factor generalization test set ($T_{OEP}$) contains 40 unseen scenario configurations that simultaneously incorporate all three types of variations: object, environment, and spatial arrangements. Second, the object generalization test set ($TO(i)$) evaluates the success rate of a robot manipulating different objects when the scenario is fixed. This test set contains all the objects introduced during the data generation process of the $i$-th iteration. A higher value of this metric not only indicates better object generalization performance of the policy but also implies a better capability to enhance the diversity of objects in data generation. All success rates reported as mean values from 5 independent runs. Detailed compositions of all evaluation sets and success rate calculation method are provided in Appendix A.4.

**Baselines.** We compared our approach against the following methods: (1) Human Demo (Default): 20 human demonstrations per task in a fixed scenario; (2) Human Demo (Enhanced): 20 demonstrations collected across diverse scenarios; (3) DexMimicGen (Default) [14]: A representative method for dexterous data generation that synthesizes trajectories via replay and editing. For fair comparison, we provide it with the same initial dataset as DexFlyWheel—a single demonstration per task. and (4) DexMimicGen (Enhanced): To create a stronger baseline, we provided DexMimicGen with 10 diverse human demonstrations collected from different scenarios. This setup significantly enhances its ability to generalize more scenarios, giving it a 10× data advantage over our method. (5) w/o Res: An ablation model featuring the base policy with $\mathcal{A}_{\text{EP}}$ but excluding the residual policy. (6) w/o $\mathcal{A}_{\text{EP}}$: An ablation model maintaining the base policy and residual policy while removing the $\mathcal{A}_{\text{EP}}$ module. (7) w/o Res. + w/o $\mathcal{A}_{\text{EP}}$: The minimal baseline configuration consisting solely of the base policy. All methods are evaluated under identical conditions: simulation setups, model architectures, and test sets, ensuring a fair and rigorous comparison.

## 5.2 Validating the Dexterous Data Flywheel Effect

This section empirically investigates **Q1**. We demonstrate how DexFlyWheel progressively expands dataset diversity and enhances policies performance trained on the generated data. As shown in the mid columns of Table 1, DexFlyWheel successfully expands data diversity with each iteration. In the final iteration (i=3), our method generates an average of 2,040 various scenario configurations spanning 20 different objects per task— *all starting from just a single human demonstration per task.* Furthermore, the object diversity results (shown in the $O$ column of Table 1) demonstrate our framework's capacity for object-level data generation. As shown in the SR of $\pi_{\text{combined}}$ on $T_{OEP}$ column, we observe that as dataset diversity increases across iterations, the generalization capabilities of trained policies correspondingly improve, achieving an average success rate of 81.9% in iteration 3—a substantial improvement from the initial 16.5% in iteration 1. Additionally, as shown in the SR Boost with $\pi_{\text{res}}$ on $T_O(i)$ column, our residual policies consistently improve performance on object generalization by 32.1% on average. These results suggest promising potential for DexFlyWheel in enhancing the data diversity and policy performance across different dexterous tasks. More detail with extended iterations can be found in Appendix A.5.

## 5.3 Comparison of Policy Performance and Data Generation Efficiency

This section evaluates DexFlyWheel and baselines in both policy performance and data generation efficiency to address **Q2**.

**Policy Performance.** We use identical diffusion-based policy architectures (Appendix A.1) and train them on datasets collected from DexFlyWheel and four baselines introduced in Section 5.1. As shown in Table 2, DexFlyWheel consistently achieves higher success rates than both human teleoperation-based data and replay-based methods such as DexMimicGen. This performance highlights the benefit of our iterative data flywheel mechanism, which progressively expands data diversity with policy improvement. Compared to the Human Demo baseline, which uses 20 teleoperated trajectories per task, DexFlyWheel achieves vastly superior performance—81.9% vs. 13.4% average success—while requiring only a single human demonstration per task, significantly reducing the human effort.

**Data Generation Success Rate and Time Efficiency.** As shown in Table 3, DexFlyWheel achieves high success across all tasks (avg. 89.8%). In contrast, DexMimicGen performs worse on dynamic

Table 1: **Self-improving Data Generation Process.** *Left)* Evaluated tasks and iteration settings. *Middle)* Dataset diversity statistics. *Right)* Success Rate (SR) of policies trained on our datasets.

| Settings | | Data Diversity | | | | | Policy Performance | |
|---|---|---|---|---|---|---|---|---|
| Task | Iter. | O ↑ | E ↑ | P ↑ | Configs ↑ | Traj. ↑ | SR Boost with $\pi_{\mathrm{res}}$ on $T_O(i)$ ↑ | SR of $\pi_{\mathrm{combined}}$ on $T_{OEP}$ ↑ |
| **Grasp** | $i=1$ | 1 | 3 | 5 | 15 | 20 | — | $15.0\%_{\pm 2.1\%}$ |
| | $i=2$ | 11 | 8 | 10 | 880 | 100 | $71.0\%_{\pm 4.3\%} \rightarrow \mathbf{84.0\%}_{\pm 3.5\%}$ | $58.0\%_{\pm 4.8\%}$ |
| | $i=3$ | **22** | **12** | **15** | **3960** | **500** | $35.0\%_{\pm 5.2\%} \rightarrow \mathbf{89.1\%}_{\pm 3.9\%}$ | $\mathbf{90.0\%}_{\pm 3.2\%}$ |
| **Pour** | $i=1$ | 1 | 7 | 3 | 21 | 20 | — | $36.1\%_{\pm 3.3\%}$ |
| | $i=2$ | 4 | 9 | 8 | 288 | 100 | $58.3\%_{\pm 5.1\%} \rightarrow \mathbf{75.0\%}_{\pm 4.0\%}$ | $55.6\%_{\pm 4.5\%}$ |
| | $i=3$ | **12** | **12** | **10** | **1440** | **500** | $58.0\%_{\pm 4.8\%} \rightarrow \mathbf{80.7\%}_{\pm 3.7\%}$ | $\mathbf{85.8\%}_{\pm 3.5\%}$ |
| **Lift** | $i=1$ | 1 | 1 | 1 | 1 | 20 | — | $13.9\%_{\pm 2.8\%}$ |
| | $i=2$ | 6 | 5 | 2 | 60 | 100 | $50.0\%_{\pm 5.3\%} \rightarrow \mathbf{83.3\%}_{\pm 3.8\%}$ | $44.4\%_{\pm 4.6\%}$ |
| | $i=3$ | **26** | **12** | **5** | **1560** | **500** | $68.8\%_{\pm 4.4\%} \rightarrow \mathbf{98.0\%}_{\pm 2.1\%}$ | $\mathbf{79.4\%}_{\pm 7.9\%}$ |
| **Handover** | $i=1$ | 1 | 1 | 1 | 1 | 20 | — | $0.8\%_{\pm 1.1\%}$ |
| | $i=2$ | 6 | 5 | 2 | 60 | 100 | $28.6\%_{\pm 5.8\%} \rightarrow \mathbf{85.7\%}_{\pm 4.2\%}$ | $17.5\%_{\pm 3.4\%}$ |
| | $i=3$ | **20** | **12** | **5** | **1200** | **500** | $32.1\%_{\pm 5.5\%} \rightarrow \mathbf{62.5\%}_{\pm 4.3\%}$ | $\mathbf{72.5\%}_{\pm 4.1\%}$ |
| **Avg.** $i=1$ | | 1.0 | 3.0 | 2.5 | 9.5 | 20 | — | 16.5% |
| **Avg.** $i=2$ | | 6.8 | 6.8 | 5.5 | 322.0 | 100 | $52.0\% \rightarrow 82.0\%$ | 43.9% |
| **Avg.** $i=3$ | | **20.0** | **12.0** | **8.8** | **2040.0** | **500** | $48.5\% \rightarrow 82.6\%$ | **81.9%** |
| **Improvement** ($i=1 \rightarrow 3$) | | **20.0×** | **4.0×** | **3.5×** | **214.7×** | **25.0×** | — | **+396.4%** |

*Notes:* **O**: Number of objects, **E**: Number of environments, **P**: Number of poses. **Configs**: total scenario configurations ($O \times E \times P$). **Traj.**: generated trajectories. **Bold** denotes the final generated datasets at the last iteration ($i=3$) and the best SR performance achieved. **Blue** indicates SR improvements from the residual policy $\pi_{\mathrm{res}}$ (i.e., $\pi_{\mathrm{base}} \rightarrow \pi_{\mathrm{combined}}$) on the object test set $T_O(i)$. **Pink** indicates generalization of $\pi_{\mathrm{combined}}$ to unseen object–environment–pose combinations in $T_{OEP}$.

Table 2: **Comparison with Baselines.** Success rates of policies trained on datasets generated by different methods when tested on multi-factor generalization test set ($T_{OEP}$).

| Method | Grasp | Pour | Lift | Handover | Avg. |
|---|---|---|---|---|---|
| Human Demo (Default) | $6.1\%_{\pm 1.2\%}$ | $16.7\%_{\pm 2.5\%}$ | $13.9\%_{\pm 2.1\%}$ | $0.8\%_{\pm 1.1\%}$ | 9.4% |
| Human Demo (Enhanced) | $15.0\%_{\pm 2.1\%}$ | $36.1\%_{\pm 3.3\%}$ | $2.5\%_{\pm 1.1\%}$ | $0\%_{\pm 0.0\%}$ | 13.4% |
| DexMimicGen (Default) | $30.3\%_{\pm 3.8\%}$ | $38.9\%_{\pm 4.2\%}$ | $28.2\%_{\pm 3.5\%}$ | $28.3\%_{\pm 4.7\%}$ | 31.4% |
| DexMimicGen (Enhanced) | $50.3\%_{\pm 4.5\%}$ | $44.4\%_{\pm 3.8\%}$ | $43.7\%_{\pm 3.6\%}$ | $42.5\%_{\pm 4.9\%}$ | 45.2% |
| Ours | $\mathbf{90.0\%}_{\pm 3.2\%}$ | $\mathbf{85.8\%}_{\pm 3.5\%}$ | $\mathbf{79.4\%}_{\pm 7.9\%}$ | $\mathbf{72.5\%}_{\pm 4.1\%}$ | **81.9%** |

tasks like Handover (14.8%). DexFlyWheel ensures robust data generation due to its policy-in-the-loop design and continuous self-improvement. For data collection time, we evaluate data collection time for each method under the most challenging setting at the final iteration ($i=3$) on Lift task. As shown in Table 4, DexFlyWheel requires only 15s per trajectory. Collecting 500 successful trajectories takes 2.4 hours—1.83× faster than DexMimicGen and over 5.21× faster than human teleoperation. See Appendix A.6 for more time details.

## 5.4 Ablation Study on DexFlyWheel Components

This section empirically investigates **Q3**. We conduct an ablation study across four manipulation tasks to isolate the impact of key modules. As shown in Figure 4, removing the residual policy leads to the most significant drop in task success rates, confirming its critical role in improving generalization and robustness. To further analysis DexFlyWheel's generalization ability, we compare the number of distinct objects successfully manipulated during data generation (Figure 5). This figure demonstrates that DexFlyWheel achieves superior object diversity compared to DexMimicGen. This performance can be attributed primarily to the residual reinforcement learning module, as evidenced by the significant drop in performance when this component is removed (w/o Res vs. ours: from 8.25 to 20 objects on average). In contrast, DexMimicGen typically operates effectively only on geometrically similar objects (same categories and shapes), which limits their generalization due to its lack of adaptability and inability to explore new strategies.

Table 3: **Data Generation Success Rate.** Success rates of generating successful demonstrations using different methods across tasks.

| Method | Grasp | Pour | Lift | Handover | Avg. |
|---|---|---|---|---|---|
| DexMimicGen | 87.3% | 81.5% | 68.2% | 14.8% | 63.0% |
| DexFlyWheel (Ours) | **93.6%** | **90.2%** | **89.5%** | **85.7%** | **89.8%** |

Table 4: **Data Generation Time.** Comparison of per-trajectory generation time and the total time required to collect 500 successful trajectories (single RTX 4090 GPU).

| Method | Time per Trajectory | Time for 500 Successful Trajectories |
|---|---|---|
| Human Teleoperation | 60s | 12.5 h |
| DexMimicGen | 15s | 4.4 h |
| DexFlyWheel (Ours) | 15s | **2.4 h** |

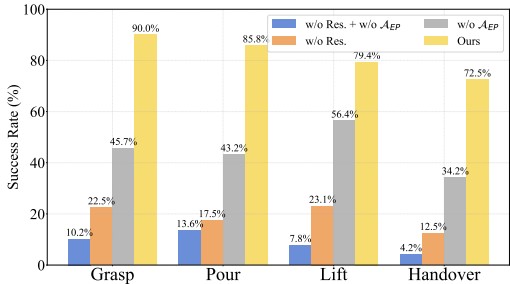

Figure 4: **Ablation Study.** Quantitative contribution of each module in DexFlyWheel across four manipulation tasks.

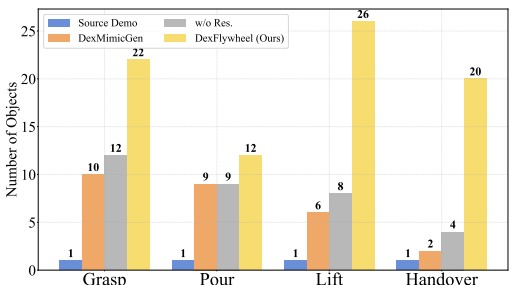

Figure 5: **Comparison of Object Diversity.** Our method successfully handles objects with diverse geometries, sizes, and categories.

## 5.5 Deployment on Dual-arm Real Robot System

To address **Q4**, we transfer our trained policy in simulation into real-world scenarios through digital twin. We set up identical hardware settings both in simulation and real-world as shown in Figure 3, which includes two Realman RM75-6F arms paired with two PsiBot G0-R hands. We employ an egocentric-view RealSense D455 camera to obtain the real-world object pose leveraging FoundationPose [61]. In particular, we train the policy using the dataset generated by the final iteration of DexFlyWheel and deploy it in the real world through the digital twin, ensuring consistency between simulation and physical environments. We evaluate the performance of our pipeline on the Dual-arm Lift and Handover Task. Experimental results show success rates of 78.3% (Dual-arm Lift) and 63.3% (Handover) in real-world deployment (20 trajectories per trial, 3 trials).

## 6 Limitations and Future Work

There are several limitations to our work. First, the reinforcement learning process currently relies on manually designed reward functions. Future research could investigate how LLM-driven reward generation methods can be efficiently integrated into DexFlyWheel. Second, our policies and simulations currently lack tactile feedback due to the immaturity of tactile sensing and simulation technologies. We plan to explore the potential of sensor-based tactile signals for contact-rich tasks.

## 7 Conclusion

We present DexFlyWheel, a scalable and self-improving framework for generating diverse, high-quality dexterous manipulation data from minimal seed demonstrations. Our two-stage pipeline first leverages imitation learning to provide behavioral priors, then applies residual reinforcement learning to enhance generalization and robustness. This approach progressively expands the data distribution across diverse objects, environments, and spatial layouts. Experiments demonstrate that DexFlyWheel can generate up to 500× more trajectories and 214× more distinct scenarios per task. Policies trained on this data achieve an 81.9% success rate in challenging settings, outperforming baselines and successfully transferring to real-world robots.

## Acknowledgements

We thank the reviewers for their constructive feedback and our labmates for insightful discussions.

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

# Supplementary Material

## Table of Contents

# A Technical Appendices and Supplementary Material

**Overview**  The Appendix contains the following content:

1. **Base Policy Implementation Details** (*Section A.1*): Details the implementation of the base policy, including model inputs and outputs, and training hyperparameters.

2. **Residual Policy Implement Details** (*Section A.2*): Describes residual policy implement details and the reward function design for the residual policy across different tasks.

3. **Data Collection and Environment Setup** (*Section A.3*): Outlines the data generation strategy incorporating environment, object, and spatial variations.

4. **Evaluation Test Set and Success Rate Calculation Method** (*Section A.4*): Presents the evaluation test set and the success rate calculation method.

5. **Performance Scaling and Extended Iteration Analysis** (*Section A.5*): Provides additional experiments and analysis to examine how performance scales with the number of flywheel iterations, and investigates whether performance improvements exhibit diminishing returns.

6. **Time Efficiency Comparison** (*Section A.6*): Compares the wall-clock time efficiency of DexFly-Wheel against baselines, highlighting both training and data generation overheads.

7. **The Minor Adjustment Observation and Curriculum Learning Strategy** (*Section A.7*): Provides the rationale for why DexFlyWheel can generalize to novel objects, describes the curriculum-based generalization strategy and our key observation, presents experimental validations across object mass and shape variations, and discusses the scope and limitations of this approach.

### A.1 Base Policy Implementation Details

This section details our base policy implementation, including model inputs and outputs, training hyperparameters and computing resources.

**Model Inputs and Outputs.** The base policy input state is denoted as $s_t = \{s_t^{\text{vis}}, s_t^{\text{obj}}, s_t^{\text{prop}}\}$, where:

Visual Input $s_t^{\text{vis}}$: For single-arm tasks, the input is a front view image $I_t^f \in \mathbb{R}^{224 \times 224 \times 3}$. For dual-arm tasks, the input is a top view image $I_t^t \in \mathbb{R}^{224 \times 224 \times 3}$.

Object State $s_t^{\text{obj}}$: In most tasks, the object state is represented by a 13-dimensional vector representing the state of a single manipulated object. For pour tasks, two objects are involved, and the object state is represented by a 26-dimensional vector.

Robot Proprioception $s_t^{\text{prop}}$: For single-arm tasks, the proprioception is $s_t^{\text{prop,single-arm}} \in \mathbb{R}^{69}$, including joint positions (19 dimensions), joint velocities (19 dimensions), gripper state (12 dimensions), gripper velocity (12 dimensions), end-effector position (3 dimensions), end-effector orientation (4 dimensions). For dual-arm tasks, the proprioception is $s_t^{\text{prop,dual-arm}} \in \mathbb{R}^{130}$, including joint positions (36 dimensions, 18 per arm), joint velocities (36 dimensions, 18 per arm), gripper states (11 dimensions per gripper), gripper velocities (11 dimensions per gripper), end-effector positions (3 dimensions per arm), and end-effector orientations (4 dimensions per arm).

**Output.** The action sequence is denoted as $d = (a_t, a_{t+1}, \ldots, a_{t+H})$ where $H = 8$. Each individual action $a_t$ includes: An end-effector 6D pose $a_t^{\text{pose}} \in \mathbb{R}^6$. Target joint angles of hands $a_t^{\text{joint}} \in \mathbb{R}^n$, where $n = 10$ for dual-arm tasks and $n = 7$ for single-arm tasks.

**Training Hyperparameters.** Table 5 summarizes all hyperparameter for the base policy training.

**Computing Resources.** All experiments are conducted on 8 NVIDIA A100 GPUs.

### A.2 Residual Policy Implement Details

This section details our residual policy implement details, including policy training and reward design.

**Policy Training.** We employ the Soft Actor-Critic (SAC) algorithm [62] to train a residual policy that enhances a pre-trained diffusion-based manipulation policy. The residual approach enables efficient learning by leveraging an existing base policy while exploring additional action refinements. Detailed hyperparameters are provided in Table 6. The residual actor network is implemented as a policy decorator that outputs corrections to the base policy's actions, allowing for fine-tuning of manipulation behaviors while maintaining the fundamental skills encoded in the base policy.

To ensure effective learning, we implement a progressive exploration strategy that gradually introduces the residual policy's influence over time. For the first 1,500 timesteps, only the base policy's actions are executed. Between 1,500 and 10,000 timesteps, the probability of including residual actions increases linearly with the global step count, promoting smooth exploration of the action space. All residual actions are scaled by a factor of 0.1 to maintain stability while allowing for meaningful corrections to the base policy. The training architecture features dual soft Q-networks with target networks updated at a rate of $\tau = 0.01$ to provide stable value estimation. The entropy coefficient $\alpha$ is automatically tuned to maintain a target entropy based on the action space dimension, balancing exploration and exploitation. Gradient updates are performed after every 5 environment steps with an updates-to-data ratio of 0.2, resulting in a total of 1 gradient update per environment step. Gradients are clipped with a maximum norm of 10 to prevent unstable updates. The critic networks evaluate the combined actions to assess the overall quality of the agent's behavior, while the actor network operates only on the proprioceptive and object state observations to generate residual corrections. This design allows the residual policy to focus on improving specific aspects of the manipulation task without requiring complete knowledge of the base policy's inner workings. The training process continues for 1.5 million timesteps, with model checkpoints saved every 10 episodes to track progress and enable resumption of training if needed.

Table 5: Hyperparameters for Diffusion Policy Training.

| Category | Parameter | Value |
|---|---|---|
| *General* | Action Steps | 8 |
| | Observation Steps | 1 |
| | Embedding Dimension | 768 |
| *Network* | Transformer Layers | 7 |
| | Attention Heads | 8 |
| | Attention Dropout | 0.1 |
| *Vision Encoder* | Model Architecture | vit_small_r26_s32_224 |
| | Pretrained | True |
| | Frozen | False |
| *Diffusion Model* | Noise Scheduler | DDIMScheduler |
| | Train Timesteps | 50 |
| | Inference Steps | 16 |
| *Training* | Batch Size | 256 |
| | Epochs | 200000 |
| | Learning Rate | 3.0e-4 |
| *Optimization* | Weight Decay | 1.0e-6 |
| | LR Scheduler | cosine |

Table 6: Hyperparameters for SAC Residual Policy Training

| Category | Parameter | Value |
|---|---|---|
| *Network Architecture* | Actor Network (MLP Layers) | [256, 256, 256] |
| | Critic Network (MLP Layers) | [256, 256, 256] |
| | State Dimension | 143 |
| | Action Dimension | 34 |
| *Training Parameters* | Learning Rate | $1.0 \times 10^{-4}$ |
| | Discount Factor ($\gamma$) | 0.97 |
| | Tau ($\tau$) | 0.01 |
| | Entropy Coefficient ($\alpha$) | 0.2 |
| | Total Timesteps | 1,500,000 |
| | Batch Size | 1024 |
| | Updates to Data Ratio | 0.2 |
| | Learning Starts | 300 |
| | Training Frequency | 5 |
| | Policy Update Frequency | 1 |
| | Target Update Frequency | 1 |
| | Max Gradient Norm | 10 |
| *Residual Strategy* | Residual Scale | 0.1 |
| | Progressive Exploration | 10,000 |
| | Progressive Exploration Threshold | 1,500 |

**Reward Design.** We carefully design the reward functions to guide the robotic manipulation policies through complex tasks. The reward functions for each task are as follows:

**Grasp Task.** The reward function for the grasping task encourages precise finger positioning and successful object lifting:

$$r_{\text{grasp}} = \exp\left(-5 \cdot \max\left(\sum_i d_i - 0.05, 0\right)\right) + 100 \cdot \max\left(0.2 - |z_{\text{target}} - z_{\text{current}}|, -0.01\right), \quad (2)$$

where $d_i$ is the distance from the $i$-th finger (thumb, index, middle) to the object center, $z_{\text{target}} = z_{\text{start}} + 0.2$ is the target height, and $z_{\text{current}}$ is the current object height.

**Pour Task.** The reward function for the pouring task guides the robot through grasping, lifting, and pouring:

$$r_{\text{pour}} = 5.0 \cdot \mathbb{I}(\text{task success}) + 10 \cdot (r_{\text{grasp\_dist}} + r_{\text{lift}}) + 50 \cdot (r_{\text{tilt}} + r_{\text{ball\_bowl}}), \quad (3)$$

where:

- $r_{\text{grasp\_dist}} = 0.5 \cdot \frac{\exp(-8.0 \cdot d_{\text{thumb}}) + \exp(-8.0 \cdot d_{\text{finger}})}{2}$,
- $r_{\text{lift}} = 50 \cdot \max(0.08 - |h_{\text{current}} - 0.08|, -0.01)$,
- $r_{\text{tilt}} = 0.5 \cdot (1 - \hat{z}_{\text{cup}} \cdot \hat{z}_{\text{up}})$,
- $r_{\text{ball\_bowl}} = 10 \cdot \exp(-5.0 \cdot \max(d_{\text{ball\_bowl}} - 0.02, 0))$.

**Lift Task.** The reward function for the lift task encourages coordinated grasping and lifting, combining the following components:

$$r_{\text{lift}} = r_{\text{left\_grasp}} + r_{\text{right\_grasp}} + r_{\text{sync}} + r_{\text{lift\_height}} - p_\theta, \quad (4)$$

where:

- $r_{\text{sync}} = 4 \cdot \exp(-5 \cdot \max(s_{\text{sync}} - 0.2, 0))$: Coordination reward based on the sum of average finger distances $s_{\text{sync}} = d_{\text{left}} + d_{\text{right}}$.
- $r_{\text{lift\_height}} = 10 \cdot \min\left(\max\left(\frac{\Delta z}{0.15}, 0\right), 1\right)$: Reward for lifting the object, where $\Delta z$ is the change in object height (target: 0.15 m).
- $p_\theta = \min\left(5.0, \frac{\theta_{\text{max}} - 30.0}{5.0}\right) \cdot \mathbb{I}(\theta_{\text{max}} > 30.0)$: Penalty for excessive tilt (threshold = 30°).
- $r_{\text{left\_grasp}}$: Reward for left-hand grasping, based on the average distance $d_{\text{left}}$ between the left fingers and the object ($\exp(-8 \cdot \max(d_{\text{left}} - 0.08, 0))$).
- $r_{\text{right\_grasp}}$: Reward for right-hand grasping, based on the average distance $d_{\text{right}}$ between the right fingers and the object ($\exp(-8 \cdot \max(d_{\text{right}} - 0.08, 0))$).

**Computing Resources.** All experiments in residual policy training are conducted on a single NVIDIA RTX 4090 GPU.

## A.3 Data Collection and Environment Setup

This section details our progressive and controlled data collection strategy for generating diverse simulation scenarios. The strategy is structured as follows:

**Progressive Data Collection Strategy.** We employ a systematic approach to cover environment variations, object variations, and spatial variations in our simulation:

- **Environment Variations**: We randomly sample environments from the available set to introduce diversity in background and lighting conditions settings.
- **Object Variations**: We adopt a curriculum-based approach, starting with geometrically similar objects and gradually introducing more challenging ones to ensure a smooth learning curve.
- **Spatial Variations**: We begin generating spatial configurations near the source demonstration scene and progressively extend them to more distant configurations within the manipulation workspace.

Each iteration of the data generation process covers a broader range of variants and presents a higher difficulty level compared to the previous one.

**Scenario Sampling Strategy.** To ensure comprehensive coverage of generalization factors (i.e., different objects, environments, spatial configurations), we design a scenario sampler. The sampler randomly samples scenarios from the entire set while guaranteeing that all factors are represented. For example, in the third iteration of the pouring task, we sample from 1440 scenarios, and the sampler selects 125 scenarios that cover 12 objects, 12 environments, and 10 spatial configurations. The scenarios are centered around objects, with different environments and spatial combinations.

**Trajectory Collection Strategy.** We set the number of iterations to $i = \{1, 2, 3\}$. For each task, we generate 20, 100, and 500 trajectories in the three iterations, respectively. Each scenario is used to collect 4 trajectories. We employ a finite mode for data collection, where in each scenario, we set the Try Time to 10 and the Success Threshold to 4 successful trajectories. If the try time exceeds 10 and the number of successful trajectories is less than 4, the scenario is flagged as failed, and we move to the next scenario. After collection, we downsample to the target number of trajectories.

## A.4 Evaluation Test Set and Success Rate Calculation Method

**Evaluation Test Set.** Our evaluation test set consists of two categories for each task:

- $T_O(i)$: The object generalization test set for each round $i$. This set is designed to evaluate the model's performance on specific objects in a given round.

- $T_{OEP}$: The comprehensive test set for each task, which includes all objects from the $T_O(i)$ sets across all rounds. The specific environments and spatial configurations for $T_{OEP}$ are detailed in the supplementary material (see the code provided).

For the $T_O(i)$ test sets, we provide visualizations for each task to illustrate the object generalization scenarios. Below are the figures for each task: Figure 6, Figure 7, Figure 8, and Figure 9.

**Success Rate Calculation Method.** (1)Grasp Task. The success condition for the grasp task is defined as: the target object must be lifted to a height exceeding 20 cm. (2)Pour Task. The success condition for the pour task is determined by checking if any ball is inside the bowl. The key evaluation metric is: a ball is considered inside the bowl if its horizontal distance to the bowl is less than 2 cm. (3) Lift Task. The success condition for the lift task is defined as: the target object must be lifted to a height exceeding 15 cm. (4) Handover Task. The success condition for the handover task is defined as: the target object must be lifted to a height exceeding 15 cm, with the right hand completely released (distance > 15 cm) and the left hand maintaining a secure grasp (distance < 10 cm) for 10 consecutive steps. (5) General Failure Condition. For all tasks except the handover task, if the execution time exceeds 600 steps without achieving the success condition, the task is deemed a failure. For the handover task, the maximum allowed execution steps are increased to 800.

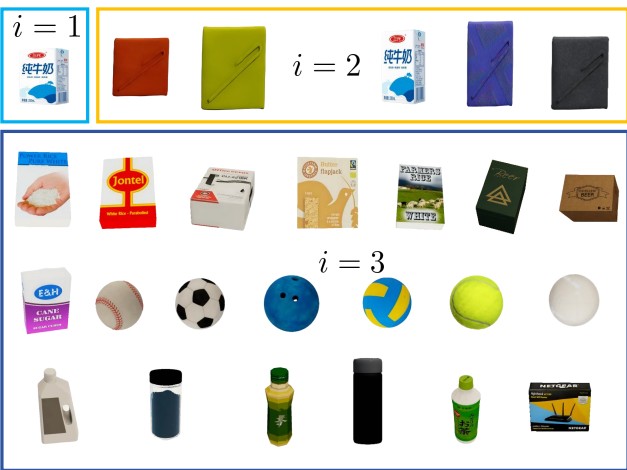

Figure 8: Lift Task Evaluation Test Set ($T_O(i)$).

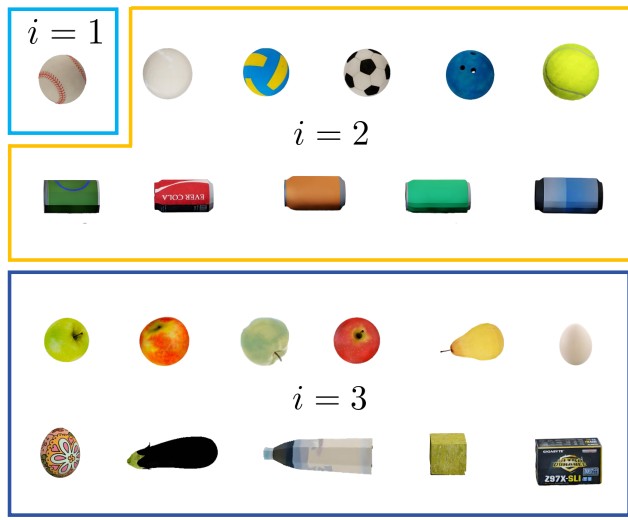

Figure 6: Grasp Task Evaluation Test Set ($T_O(i)$).

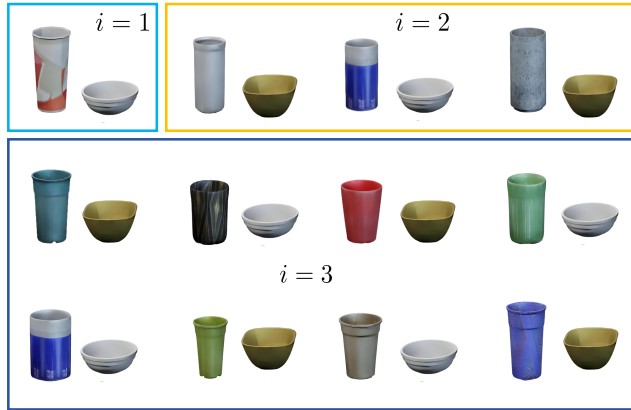

Figure 7: Pour Task Evaluation Test Set ($T_O(i)$).

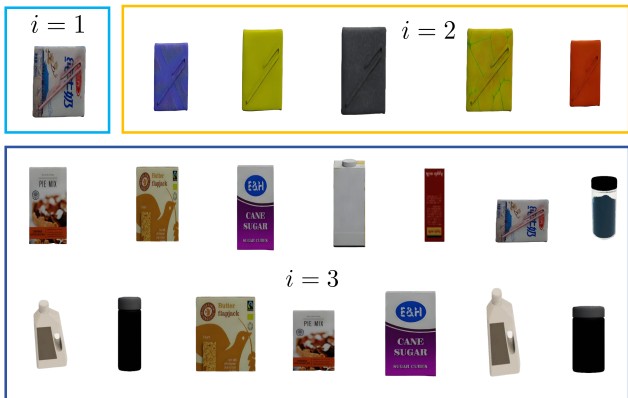

Figure 9: Handover Task Evaluation Test Set ($T_O(i)$).

## A.5 Performance Scaling and Extended Iteration Analysis

To examine the effect of further iterations, we evaluated DexFlyWheel performance up to iteration 5 on the Grasp and Lift tasks:

Table 7: **Extended iteration performance of DexFlyWheel.** Success rates of policies on Grasp and Lift tasks across iterations. Values are reported as mean ± standard deviation, with the improvement over the previous iteration.

| Iteration | Grasp SR (%) | Lift SR (%) |
|---|---|---|
| i = 1 | 15.0 ± 2.1 | 13.9 ± 2.8 |
| i = 2 | 58.0 ± 4.8 (+43.0) | 44.4 ± 4.6 (+30.5) |
| i = 3 | 90.0 ± 3.2 (+32.0) | 79.4 ± 7.9 (+35.0) |
| i = 4 | 92.5 ± 2.8 (+2.5) | 82.1 ± 6.5 (+2.7) |
| i = 5 | 93.2 ± 2.5 (+0.7) | 83.5 ± 5.8 (+1.4) |

These results indicate that iteration $i = 3$ provides a practical trade-off between performance gain and computational cost. Performance continues to improve in later iterations, demonstrating the potential of DexFlyWheel to further enhance data diversity with additional iterations.

## A.6  Time Efficiency Comparison

**Wall-clock Training Time.** Table 8 reports the wall-clock time for training the DexFlyWheel policies. The base policy trained with IL requires 5h 40m, while the residual RL policy requires 6h 30m per iteration. Across the full three-iteration DexFlyWheel process, total wall-clock training time is approximately 30 hours. Note that the first iteration uses only IL.

Table 8: **Wall-clock Training Time.** Wall-clock time required to train the base policy and residual policies for three DexFlyWheel iterations.

| Policy Training (Iteration) | Wall-clock Time |
|---|---|
| Base policy (IL) | 5h 40m |
| Residual policy (RL) | 6h 30m |
| Total (3 iterations) | 30 hours |

## A.7  The Minor Adjustment Observation and Curriculum Learning Strategy

DexFlyWheel generalizes effectively to novel objects by leveraging the *Minor Adjustment Observation* and a curriculum-based policy learning strategy. As introduced in the main text, we observe that manipulating different objects typically causes only minor changes in the manipulation trajectories. In this appendix, we first present experimental evidence supporting this observation and . Based on these insights, we then describe our curriculum-based policy training strategy, which progressively exposes the policy to more diverse objects to enhance generalization.

### A.7.1  Experimental Validation of the Minor Adjustment Observation

**Experimental Setup.**   To rigorously quantify the effect of object variations on manipulation trajectories, we conducted controlled experiments along two axes: (1) varying object mass in a dual-arm lifting task, and (2) varying object shape in a grasping task. Trajectory deviation is measured using **JointDiff** (mean absolute joint position difference in radians between the nominal trajectory and the adapted trajectory). We also track task success rates across iterations ($i = 1, 2, 3$). To evaluate the subtlety of adjustments, we define the **Residual Norm Ratio (RNR)**:

$$\text{RNR} = \frac{\|a_t^{\text{res}}\|}{\|a_t^{\text{base}}\| + \epsilon},$$

where $a_t^{\text{res}}$ is the residual correction at timestep $t$, $a_t^{\text{base}}$ is the base action, and $\epsilon = 10^{-6}$ prevents division by zero. Low RNR indicates minor adjustments rather than drastic changes.

**Key Findings.**   Our experimental results provide strong evidence for the Minor Adjustment Observation and the effectiveness of our curriculum strategy:

Table 9: **Trajectory Difference under Varying Mass.** JointDiff (radians) when the object mass is varied in a dual-arm Lift task.

| Objects (Density) | JointDiff (rad) |
|---|---|
| Very Light Box (0.1) | 0.23 |
| Original Box (1) | – |
| Heavy Box (10) | 0.74 |
| Very Heavy Box (50) | 1.06 |

Table 10: **Trajectory Difference under Varying Shape.** JointDiff (radians) when object shape is varied in a Grasp task.

| Objects | JointDiff (rad) |
|---|---|
| Tennis Ball (Default) | – |
| Bowling Ball | 0.75 |
| Coke Can | 0.59 |
| Eggplant | 0.95 |
| Water Bottle | 1.18 |

Table 11: **Policy Success Rates and Residual Norm Ratios.** Success rates (SR) across curriculum iterations and average RNR for different objects.

| Objects | SR $i = 1$ | SR $i = 2$ | SR $i = 3$ |
|---|---|---|---|
| Very Light Box (0.1) | 86.7 | 92.0 | 93.3 |
| Heavy Box (10) | 36.7 | 45.0 | 80.5 |
| Very Heavy Box (50) | 0.0 | 12.0 | 56.0 |
| Tennis Ball (Default) | 94.0 | 93.7 | 94.2 |
| Bowling Ball | 43.1 | 78.9 | 90.0 |
| Coke Can | 38.0 | 73.4 | 80.2 |
| Eggplant | 20.6 | 30.8 | 70.9 |
| Cube | 15.9 | 20.0 | 85.6 |
| Average RNR (%) | – | $14.3 \pm 2.6$ | $16.5 \pm 2.1$ |

- **Trajectory Adjustment Remain Modest:** As shown in Tables 9 and 10, trajectory adjustment (JointDiff) remain relatively modest under reasonable object variations. While differences grow larger for extreme object properties (e.g., density 50.0 or highly non-spherical shapes like a water bottle), they generally indicate an adaptation rather than a complete replanning of the trajectory.

- **Curriculum-Based Training Substantially Improves Success Rates:** Table 11 clearly demonstrates that our curriculum-based training strategy substantially improves success rates across all tested conditions, especially for initially challenging cases (e.g., eggplant, cube, very heavy objects). After three iterations, DexFlyWheel achieves strong performance even on objects that yielded very low success rates in early iterations. This highlights how the curriculum effectively guides the learning process to generalize to novel objects.

- **Low Residual Norm Ratio Confirms Minor Adjustments:** The relatively low Average Residual Norm Ratio values (in the order of $10^{-2}$) confirm that the residual corrections generated by DexFlyWheel are indeed subtle adjustments to the base actions. This directly supports the Minor Adjustment Observation, indicating that the policy learns to finely adapt pre-existing trajectories rather than generating entirely new ones from scratch for novel objects.

**Scope and Limitations.** The Minor Adjustment Observation holds for tasks where fundamental interaction modes remain consistent (e.g., grasping, lifting, pouring, handover). It may not generalize to tasks requiring drastically different strategies, such as deformable object manipulation (e.g., cloth folding, knot tying) or precision assembly, where subtle changes in object properties may require fundamentally different control strategies.

### A.7.2 Curriculum-Based Policy Learning Strategy

Based on the Minor Adjustment Observation, we design a curriculum-based policy training strategy to progressively generalize to novel objects. Using the dual-arm lift task as an example:

- **Iteration 1: Foundational Skills with a Simple Object.** In the initial iteration ($i = 1$), we begin training with a simple object. This foundational step allows the robot to acquire basic grasping and lifting skills in a simplified environment, establishing a robust baseline for subsequent learning.

- **Iteration 2: Generalization to Geometry-Similar Objects.** Following the foundational stage ($i = 2$), we introduce a set of objects that share geometric similarities with the initial training object (e.g., objects of similar primitive shapes but varying dimensions). By training on these geometry-similar objects, the robot starts to generalize its skills beyond the exact specifications of the simple box. During test-time evaluation, we observe that DexFlyWheel not only performs well on these seen objects but also demonstrates an initial level of generalization to unseen objects with different geometries within this category.

- **Iteration 3: Generalization Across Diverse Object Categories.** In the third iteration ($i = 3$), we expand the object diversity by incorporating various categories with different geometries and appearances. DexFlyWheel demonstrates improved generalization compared to earlier iterations, performing reliably on a wider range of previously unseen objects. This highlights the effectiveness of the curriculum-based strategy in supporting the robot's ability to handle diverse object types.

