# OpenReview forum: "DexFlyWheel: A Scalable and Self-improving Data Generation Framework for Dexterous Manipulation"
_NeurIPS.cc/2025/Conference — NeurIPS 2025 spotlight_

### Official Review · Reviewer_XGgH · 2025-06-25

**Clarity:** 3
**Significance:** 3
**Originality:** 2
**Rating:** 4
**Confidence:** 4

**Summary:**

This paper proposes a data flywheel approach to training robots in simulation. A VR teleoperation interface collects a small set of human demonstrations that are used to train a behavior cloning policy. This base policy is then modified with a residual policy using reinforcement learning in a simulated setup. Data collected with this modified policy is filtered, augmented, and used to improve the base policy using imitation learning. Results show iterative improvement as cycles of the flywheel are run, including transfer to real robot setups.

**Questions:**

- If DexFlyWheel requires simulators and rewards, how is this different and better than other RL at scale works like MT-Opt (https://arxiv.org/pdf/2104.08212) that also have a filtered data loop?
- Why are initial success rates at iter i = 1 (Table 2) higher than the initial SR in iters i = 2 and i = 3? The near 100% success rates feel odd given that the policy has only been trained on 20 trajectories, especially because the robot is using  What can explain the high initial success rates and the initial ability for the robot to learn from 20 trajectories only?
- What exactly is your data augmentation module A_EP doing? It is briefly mentioned in lines 149-155 but it seems like a contribution in the pipeline, so it is worth discussing in more detail if possible.

I’d be willing to increase the scores and overall rating if the authors present a compelling argument on the originality of this approach and its advantages over other continual RL/IL paradigms, as well as clarified why TO^i results start high and decrease.

**Ethical Concerns:**

["NO or VERY MINOR ethics concerns only"]

**Final Justification:**

See response to author rebuttal for the full details. Nearly all of my concerns from the original reviews were original misunderstandings (clarified by the authors) or addressed by additional experiments. Rebuttal remarks about DexFlywheel's scope and novelty were also convincing. I would appreciate an increase of paper clarify in the final publication to dispel similar misunderstandings by other readers.

**Limitations:**

- Specify the need for simulations for this pipeline to work (given the amount of residual RL and rewards needed)
- Reduce the strength of the paper claim: “infinite diverse data generation only from one demonstration per task”

**Paper Formatting Concerns:**

no major problems

**Quality:**

3

**Strengths And Weaknesses:**

Strengths

- VR teleoperation with bimanual hand tracking makes human data collection relatively straightforward
- meaningful results with sim2real transfer
- results show improvement in generalization tests with each iteration of the flywheel

Weaknesses

- although this approach requires comparatively few human demonstrations, it also requires access to ground truth simulation reward and carefully engineered reward functions
- Requires simulation and carefully engineered rewards to work with the RL part
- The strong claim of “infinite diverse data generation only from one demonstration per task” does not match the experiments, which collect 20 trajectories per task in the first iteration. It is an overclaim.
- Performance drop in T_O^i (Table 2) with added data does not make sense and not much explanation is given about this observation. It is especially surprising given that the tasks require dexterous manipulation.
- filtering requires success detection, which is accessible in simulation but not in the real world without additional modues
- There exist a growing body of work focusing on data flywheel-type approaches, but there are limited mentions of these works. These works below may not all be fully relevant, but provided here for convenience
    - https://arxiv.org/pdf/2411.01813v1
    - https://arxiv.org/abs/2405.10315
    - https://arxiv.org/pdf/2407.16677v1
    - https://arxiv.org/pdf/2505.10442v1
    - https://proceedings.mlr.press/v164/lu22a/lu22a.pdf
    - https://arxiv.org/pdf/2410.14868v4

---

> ### Author Rebuttal · Authors · 2025-07-31
>
> # **Response to Reviewer XGgH**
> We sincerely appreciate your thoughtful summary and recognition of our work’s strengths. Below, we address the concerns you raised.
>
> ## **Clarifying Our Work’s Focus**
> While you characterized our method as a “policy training approach,” we would like to clarify that DexFlyWheel’s primary contribution is a scalable, data-centric framework for generating diverse, high-quality dexterous manipulation data, rather than a new policy learning algorithm. This distinction is crucial for understanding our work’s novelty and scope.
>
> ---
>
> ## **Q1. If DexFlyWheel requires simulators and rewards, how is it different and better than other large-scale RL works like MT-Opt that also use filtered data loops?**
>
> ### **A1. Different Goals and Lightweight Residual RL Module**
> Our goal fundamentally differs from that of RL algorithm papers such as MT-Opt. While MT-Opt focuses on multi-task RL policy design for gripper-based tasks, DexFlyWheel focuses on building a scalable data generation framework for dexterous manipulation involving high-DOF hands and complex tasks.
>
> Additionally, our residual RL module is lightweight and efficient, used to *refine* base policies trained via imitation learning (IL) to improve generalization while preserving human-like behavior. This contrasts with MT-Opt’s more complex RL modules focused on large-scale policy learning.
>
> ## **Q2/W4. Why are initial success rates at iteration i=1 (Table 2) higher than at i=2 and i=3? How can the policy perform well with only ~20 trajectories?**
>
> ### **A2. Increasing Test Set Difficulty Across Iterations**
> $T_O^i$ is not a fixed test set—each round introduces harder objects and more diverse scenes, making later evaluations progressively more challenging (Sec. 5.1, L233–237). To clarify this, we added a new experiment on the Lift task, where we evaluate policies from each iteration on all test sets ($T_O^1$: easy, $T_O^2$: medium, $T_O^3$: hard):
>
> | Task  | Iter. | $T_O^1$ (Easy) | $T_O^2$ (Medium) | $T_O^3$ (Hard) |
> |-------|-------|--------------------|---------------------|--------------------|
> | Lift  | 1     | 90.0 ± 2.2        | 45.0 ± 3.1         | 18.9 ± 2.5        |
> |       | 2     | 93.0 ± 3.5        | 83.3 ± 3.8         | 59.0 ± 6.9        |
> |       | 3     | 93.4 ± 2.6        | 90.3 ± 1.5         | 98.0 ± 2.1        |
>
> These results confirm continuous improvement on increasingly challenging test sets as the flywheel iterates.
> We have renamed $T_O^i$ to $TO(i)$ in the revised manuscript for clarity (Sec. 5.1, L233–237, Table 1).
>
> ---
>
> ## **Q3. What exactly does the data augmentation module $A_{EP}$ do? It’s briefly mentioned but seems important.**
>
> ### **A3. Multi-Level Scene Augmentation to Enrich Data Diversity**
> The  $A_{EP}$ module applies multi-level augmentations during warm-up and flywheel phases:
> - Environment-Level Enhancements:
>   - Lighting variation (e.g., ±20% brightness, color temperature shifts)
>   - Workspace appearance changes (e.g., wood to metal surfaces)
> - Spatial-Level Enhancements:
>   - Object Pose Variation to object initial poses (e.g., ±5 cm translation)
>
> This module is first applied to seed demonstrations to bootstrap a diverse distribution of initial trajectories (20 demos), and it continues to promote data diversity during the iterative flywheel phase.
>
> In the revised manuscript (Section 4.3, lines 149–151), we have updated the description of the data augmentation module $A_{EP}$ to provide a more comprehensive clarification of the above.
> ## **Q4. Comparison to continual IL/RL approaches (which in the reviewer questions summary)**
>
> ### **A4. Hybrid IL + Residual RL Framework for Efficient, Human-like and Diverse Data Generation**
> Thank you for the insightful question!  This is a critical point that gets to the heart of our core design:
> - Imitation Learning (IL) provides human-like behavior priors and significantly accelerates the exploration process in reinforcement learning.
> - Residual Reinforcement Learning (Residual RL) efficiently refines the base policy learned via IL, improving generalization and enabling diverse, high-quality data generation.
>
> In contrast:
> - Pure IL tends to suffer from poor generalization to novel objects or scenarios.
> - Pure RL often produces unnatural or low-quality trajectories and suffers from low exploration efficiency.
>
> We have conducted experiments to validate these claims.
>
> **Compared to continual pure IL:**
>
> - In our manuscript, experiments (see Fig. 5) already confirm that continual pure IL struggles to generalize to novel objects. Specifically, it achieves only 8.95% success on the challenging test set—a 72.95% drop compared to DexFlyWheel. Even with strong augmentations, the performance improves only marginally to 18.9%, highlighting the limitations of continual pure IL in generalization.
>
> **Compared to continual pure RL:**
>
> We ran an additional Lift task experiment (from iteration 2 to 3) comparing continual pure RL and our method:
>
> | Method           | SR  i=3 ↑ | Training Time ↓ | Trajectory Jerkiness ↓ |
> |------------------|-------------|-----------------|-----------------------|
> | Continual Pure RL | 43.8 ± 5.7 | 20 hours        | 2.07                  |
> | Ours             | 98.0 ± 2.1 | 6.5 hours       | 0.28                  |
>
> - Performance: Our approach achieves a significantly higher SR (98.0 vs. 43.8) due to the strong IL-based initialization.
> - Efficiency: Continual RL must explore from scratch (20h), while our method benefits from the base IL policy, reducing RL search space (6.5h).
> - Trajectory Quality: Continual Pure RL often generates jerky, non-human-like motions. Our method preserves smooth and natural behavior (0.28 vs.2.07).
>
> ## **W1/W2. Although your method needs few human demos, it requires ground truth simulation rewards and careful reward design.**
> ### **A5. Compatibility with Sparse and LLM-Driven Rewards**
>
> We demonstrate that DexFlyWheel is compatible with sparse and LLM-driven rewards (Lift task):
>
> | Reward Type           | Success Rate (%) | Training Steps    |
> |----------------------|------------------|-------------------|
> | Human-designed [1]    | 98.0 ± 2.1       | 0.5M              |
> | LLM-driven [2]       | 80.9 ± 5.7       | 1.0M              |
> | Sparse               | 84.5 ± 3.6       | 2.0M              |
>
> Many mature reward functions for common dexterous manipulation tasks have already been widely adopted in the community [1]. Applying these well-established reward formulations in DexFlyWheel proves more efficient than using sparse or LLM-driven rewards [2], due to faster convergence and more stable policy training (2~4 times fewer training steps).
>
> In future work, we plan to explore how LLM-driven reward generation methods can be efficiently integrated into DexFlyWheel.
>
> ## **W3. The claim of “infinite diverse data from one demonstration” conflicts with collecting 20 demo trajectories initially.**
> ### **A6:**
> We respectfully clarify your misunderstanding: our method relies on only a single human demonstration per task, as described in Section 4.2 (Line 146). The “20 trajectories” reported in Table 1 are automatically generated through our data augmentation module from that single demo, with no further demonstration input.
>
> ## **W5. Filtering requires success detection, which is accessible in simulation but not in the real world without additional modues.**
>
> ### **A7:**
> We would like to clarify that our approach is a simulation-based data generation framework, which does not require collecting or filtering data in the real world. This is a common design choice adopted in many representative simulation-based data generation methods [3–5].
>
> ## **W6. There exist a growing body of work focusing on data flywheel-type approaches, but there are limited mentions of these works. These works below may not all be fully relevant, but provided here for convenience**
>
> ### **A8:**
> While several recent works have explored IL/RL-based flywheel concepts, they primarily focus on gripper-based tasks and emphasize policy tuning or online correction rather than data generation itself. In contrast, DexFlyWheel is specifically designed for dexterous manipulation involving high-DOF hands and complex tasks, and it serves as a complete data-centric framework with a principled flywheel design that drives scalable data generation and policy improvement.
>
>
> ### **Revisions**
> - **Related Work:** Expand discussion of continual IL/RL and flywheel-style frameworks, clarifying DexFlyWheel’s distinct contributions.
> - **Method:** Extended the description of the $A_{EP}$ module to highlight its role in generalizing across task variations through scene randomization and trajectory replay.
> - **Notation:** Rename $T_O^i$ to $TO(i)$ for clarity.
>
> ---
>
> We appreciate your valuable feedback, which improved our paper’s clarity. Other reviewers recognize the novelty of our closed-loop flywheel, scalability (2000+ trajectories), and real-world effectiveness (78.3% dual-arm lift SR). We hope our detailed responses demonstrate DexFlyWheel’s significance as the first policy-in-the-loop data generation framework for dexterous manipulation, and respectfully request reconsideration of your rating.
>
> ---
>
>
> **References:** \
> [1] Bi-DexHands: Towards human-level bimanual dexterous manipulation IEEE Transactions on Pattern Analysis and Machine Intelligence, vol. 46, no. 5, pp. 1-15, May 2024.\
> [2] Eureka: Human-level reward design via coding large language models," in Proc. Int. Conf. Learn. Representations (ICLR), Jan. 2024.\
> [3] A. Mandlekar et al., “MimicGen: A Data Generation System for Scalable Robot Learning using Human Demonstrations,” CoRL, 2023. \
> [4] R. Wang et al., “DexGraspNet: A Large-Scale Robotic Dexterous Grasp Dataset for General Objects Based on Simulation,” ICRA, 2023. \
> [5] Z. Jiang et al., “DexMimicGen: Automated Data Generation for Bimanual Dexterous Manipulation via Imitation Learning,” ICRA, 2025.

---

> > ### Comment · Reviewer_XGgH · 2025-08-01
> > **Acknowledgement of Rebuttal & Some Misunderstandings; Score Improvement**
> >
> > Hi Authors! I really appreciate the thorough response to my concerns, concrete clarifications of my misunderstandings, and additional results. In general I was very impressed at these clarifications.
> >
> > # Rethinking Novelty Assessment
> > I appreciate the author's clarification on the novelty component of DexFlywheel. While it is true that the general idea of a data flywheel has existed for a good amount of time, this particular execution for RL on dexterous manipulation + flywheel is a meaningful contribution. A4 provides a compelling result showcasing how DexFlywheel can outperform pure RL across different metrics, which demonstrates that the reward engineering was not the part that made DexFlywheel operational. I also appreciate the scoping of DexFlywheel as a simulation-based method, which removes my concerns about easy real world transfer.
> >
> > # Clarified Result Understanding of $T_O^i$
> > Thank you for the clarification of what $T_O^i$ stands for, and the chart in A2 is a very helpful clarification. They convince me of the positive impact of this additional data. It could be nice to include a table / graph like that in A2 in the final publication to showcase this point a bit more clearly.
> >
> > # Infinite data from one demonstration
> > Thank you for this clarification, and I retract my concern about the misrepresentation of results. I would recommend in the final publication to represent the "pure" data and the augmented data in a more clear fashion on the table.
> >
> > # Dependency on reward function
> > A5 is a compelling demonstration that DexFlywheel depends less on a good reward structure than initially thought. This showcases the impact of imitation learning base policies as mentioned by the authors.
> >
> > # Summary & Changes
> > I mentioned in my review that I would increase my ratings if I got meaningful clarification on my original concerns with performance and novelty. I believe that this rebuttal has sufficiently addressed these concerns and my method misunderstandings. The final publication should address these misunderstandings as places where additional clarity is needed.
> >
> > I increase my score from *reject* to *borderline accept* and increase the scores in other categories accordingly.
> > What keeps my score from increasing further is the assumptions that keep DexFlywheel from being applied very straightforwardly to real robot tasks (e.g. needing digital twin environments and good models), although the provided real robot results are a compelling proof of concept of what could be done.

---

> ### Author Response · Authors · 2025-08-06
>
> Dear Reviewer XGgH,
>
> Thank you sincerely for your detailed re-evaluation and for engaging so constructively with our rebuttal. We are delighted that our responses have clarified your initial concerns regarding the novelty of our framework and the interpretation of our results. **We are very grateful for increasing your rating of our paper!**
>
> We also appreciate your valuable suggestions for improving the final manuscript. As you recommended, we will make the following additions to enhance clarity:
>
> **(1) Clarifying Progressive Difficulty:** We will include a table in the revised manuscript (similar to the one in our rebuttal, A2) to clearly demonstrate how the policy’s performance improves across increasingly difficult test sets $T_O^i$ with each flywheel iteration.
>
> **(2) Clarifying Data Source:** We will revise our results tables (specifically, the “Traj.” column in Table 1 of the submission manuscript) to explicitly distinguish the initial human seed demonstration from the trajectories that are automatically generated and augmented by our framework.
>
> **Regarding your remaining point** on real-world applicability and reliance on digital twin environments, thank you for recognizing our real-robot experiments as a compelling proof-of-concept. We wish to clarify that our core contribution is tackling the critical challenge of data generation in dexterous manipulation. Digital twin environments are a standard and widely adopted choice for validating data generation frameworks (e.g., DexMimicGen [1]) because they provide a safe, repeatable, and scalable testing pipeline. This choice allowed us to focus on verifying the quality and effectiveness of the generated data itself.
>
> **Additionally,** our work is orthogonal yet synergistic to sim-to-real research: sim-to-real focuses on transferring policies, while DexFlyWheel addresses the bottleneck of generating diverse, high-quality training data. Even with perfect simulators and digital twin environments, acquiring such data at scale remains a critical challenge. DexFlyWheel makes a significant step toward solving this specific challenge and lays a strong foundation for future robust sim-to-real transfer. Moreover, as simulation fidelity [2,3] and sim-to-real transfer [4,5] technologies continue to advance, the sim-to-real gap is expected to progressively narrow, which in turn will further accelerate the deployment of methods like DexFlyWheel in real-world applications.
>
> **Looking ahead,** while our current development and evaluation platform is based on simulation, the flywheel principle itself is not limited to simulation. A practical next step is to directly connect VR teleoperation to real robots to collect seed demonstrations, followed by training a base policy. Residual reinforcement learning can then be applied on the real hardware to refine performance and collect rollout data [6], with rule-based checks automatically filtering out failed executions. These successful trajectories can be further enhanced through world model [7] before proceeding to the next iteration. As dexterous robotic hardware becomes increasingly available in the real world, we believe DexFlyWheel has the potential to extend from simulation to real-world data generation for dexterous manipulation.
>
> Thank you once again for your constructive and valuable engagement. With our planned revisions and the contextualization of our contribution, we hope we have now fully addressed all of your concerns. We would be grateful for your confirmation or any final thoughts you might have.
>
> ---
>
> **References:** \
> [1] Jiang, Z. et al., "DexMimicGen: Automated Data Generation for Bimanual Dexterous Manipulation via Imitation Learning", ICRA 2025. \
> [2] Li, C., et al., "BEHAVIOR-1K: A Human-Centered, Embodied AI Benchmark with 1,000 Everyday Activities and Realistic Simulation", arXiv preprint arXiv:2403.09227, 2024. \
> [3] Tao, S., Xiang, F., Shukla, A., et al., "ManiSkill3: GPU Parallelized Robotics Simulation and Rendering for Generalizable Embodied AI", arXiv preprint arXiv:2410.00123, revised May 2025. \
> [4] R. Singh. et al., “DextrAH-RGB: Visuomotor Policies to Grasp Anything with Dexterous Hands,” arXiv preprint arXiv:2412.01791, 2025. \
> [5] Y. Jiang, C. Wang, R. Zhang, J. Wu, and L. Fei‑Fei, “TRANSIC: Sim‑to‑Real Policy Transfer by Learning from Online Correction,” arXiv preprint arXiv:2405.10315, 2024. \
> [6] J. Luo, C. Xu, J. Wu, S. Levine, “Precise and Dexterous Robotic Manipulation via Human-in-the-Loop Reinforcement Learning,” arXiv preprint arXiv:2410.21845v1, 2025. \
> [7] S. Huang et al., “EnerVerse: Envisioning Embodied Future Space for Robotics Manipulation,” arXiv preprint arXiv:2501.01895, 2025.

---

> ### Author Response · Authors · 2025-08-09
>
> Dear Reviewer XGgH,
>
> We apologize for the inconvenience of reaching out again regarding the sub-score update. Thank you very much for your response and for your decision to increase both the overall rating and the sub-scores.
>
> However, we noticed that the sub-scores currently remain the same as the initial ratings: Quality: 2 – fair; Clarity: 2 – fair; Significance: 1 – poor; Originality: 1 – poor. We are a bit unsure whether this might be due to a system issue or if the adjustment has not yet been processed.
>
> We truly appreciate your time, consideration, and efforts in reviewing our work. If you have any further questions or if there is any additional information we can provide, please feel free to let us know.
>
>
> Best regards,
> The Authors

---

### Official Review · Reviewer_5Q2M · 2025-07-03

**Clarity:** 3
**Significance:** 4
**Originality:** 3
**Rating:** 5
**Confidence:** 3

**Summary:**

The paper introduces DexFlyWheel, a scalable framework for generating high-quality and diverse dexterous manipulation datasets from minimal human demonstrations. It combines imitation learning (IL) and residual reinforcement learning (RL) in a self-improving cycle. Starting from a single human demonstration per task, DexFlyWheel iteratively augments data diversity and policy generalization by rolling out trajectories in simulation, applying domain randomization, and refining policies via RL. The method is validated on multiple dexterous manipulation tasks (grasp, pour, lift, handover) and achieves strong performance both in simulation and real-world deployment

**Questions:**

- Why were only the dual-arm tasks (Lift, Handover) deployed in real-world experiments? Were there specific challenges in deploying single-arm tasks (Grasp, Pour)? Including even preliminary hardware results for single-arm tasks would strengthen claims of broad applicability.
- The absence of real-world deployment for baseline methods limits comparative insights into sim-to-real performance. Could the authors discuss any technical or fairness challenges that prevented such evaluation? Would they consider adding such comparisons in future work or supplementary material?
- The current evaluation uses four rigid-object tabletop tasks. Can the authors comment on the feasibility of extending DexFlyWheel to tasks with articulated objects, or long-horizon manipulation? This would provide a more robust test of the framework’s scalability.

**Ethical Concerns:**

["NO or VERY MINOR ethics concerns only"]

**Final Justification:**

I have read the rebuttal the authors posted, which addressed my questions and concerns.

**Limitations:**

Yes

**Quality:**

3

**Strengths And Weaknesses:**

Strengths:
- Novelty: Introduces a closed-loop “FlyWheel” mechanism integrating IL and RL, enabling scalable data generation from minimal seed demonstrations.
- Highly Scalable: Achieves over 2000 diverse demonstrations from just one teleoperated demo per task.
- Extensive simulation experiments with multiple baselines and ablation studies.
- Demonstrates promising sim-to-real transfer on dual-arm Lift tasks.

Weakness:
- While the simulation comparisons are thorough, it would significantly strengthen the claims of superior real-world generalization if baseline methods used in simulation were also deployed on hardware. This would help quantify the sim-to-real gap across methods and ensure fair evaluation.
- While the real-world deployment of DexFlyWheel on dual-arm tasks demonstrates promising sim-to-real transfer, no real hardware results are reported for single-arm tasks (Grasp and Pour). This omission limits the ability to fully assess the framework’s generalization performance across all studied tasks.
- The framework’s evaluation covers 4 dexterous manipulation tasks, which provides initial evidence of scalability. However, these tasks are limited. It would be helpful to add more tasks.

---

> ### Author Rebuttal · Authors · 2025-07-31
>
> # **Response to Reviewer 5Q2M**
> We sincerely thank you for the thoughtful and encouraging feedback. We are glad that you recognized the novelty of our closed-loop DexFlyWheel framework, its significance for robotics community and its highly scalable for data generatoin. Below, we address your additional questions and suggestions in detail.
>
> ---
>
> ## **Q1/Q2/W1/W2. Simulation Baselines and Single-Arm Task Real-World Results in Real-world**
>
> ### **A1:**
> Thank you for pointing this out. In the submission manusricpt, we chose to showcase real-world deployment on more challenging bimanual tasks, as they better highlight the capabilities of DexFlyWheel in handling complex dexterous scenarios. As suggested, we extended our evaluation to include (1) single-arm tasks (Grasp and Pour) and (2) real-world deployments of the top-performing baseline method (DexmimicGen). To ensure a fair comparison, all methods were deployed following an identical pipeline, as described in Section 5.5.
>
> | Method       | Single-arm Grasp (%) | Single-arm Pour (%) | Dual-arm Lift (%) | Dual-arm Handover (%) | Average (%) |
> |--------------|---------------------|--------------------|-------------------|----------------------|-------------|
> | Ours         | 79.8                | 76.4               | 78.3              | 63.3                 | 74.45       |
> | DexmimicGen  | 70.5                | 35.6               | 59.1              | 10.2                 | 43.85       |
>
> The results show that our method demonstrates strong performance on both single- and dual-arm tasks, confirming its broad applicability. Furthermore, it significantly outperforms the baselines across all real-world tasks, highlighting its superior potential for practical application.
>
> **Revisions:**
> We have incorporated these new findings into the revised manuscript in Sec. 5.5, featuring a new "Table 4: Real-World Deployment Results with Baselines." These results provide direct evidence of our method's robustness and effectiveness in practical scenarios.
>
>
> Hope the above responses address your concerns. Thank you again for your time and effort!
>
> ## **Q3/W3. The current evaluation uses four rigid-object tabletop tasks. Can the authors comment on the feasibility of extending DexFlyWheel to tasks with articulated objects, or long-horizon manipulation?**
>
> ### **A2:**
> We thank the reviewer for this question regarding the scalability of our framework. Conceptually, our framework is designed with scalability in mind. It is task-general by design: the VR interface allows flexible teleoperation for collecting demonstrations across diverse tasks, and the combination of imitation learning and residual reinforcement learning operates on standard inputs (e.g., RGB images, object state, and robot state) that are applicable to a wide range of manipulation scenarios.
>
> Following your valuable suggestion, we have conducted new experiments to test the feasibility of extending DexFlyWheel to more complex scenarios, specifically articulated object manipulation (opening a door).
>
> | Task      | Iter. | O | E | P  | Configs | Traj. | SR in TOEP (%) |
> |-----------|-------|---|---|----|---------|-------|----------------|
> | Open Door | i=1   | 1 | 1 | 1  | 1       | 20    | 34.2           |
> |           | i=2   | 2 | 5 | 5  | 50      | 100   | **70.6**           |
>
> Our preliminary results from these new experiments are encouraging. They demonstrate that DexFlyWheel can effectively scale to more complex tasks.
>
> Regarding task horizon, our existing Pour and Handover tasks already exhibit a certain degree long-horizon characteristics. The Pour task requires a two-stage pick-and-pour sequence, while Handover involves coordinated bimanual manipulation, including object pickup and hand-to-hand transfer (Sec. 5.1, L208–209 & L211–212).
>
> To further evaluate long-horizon scalability, we conducted an additional three-stage task: Pick-Pour-Place, where the robot must (1) pick up a container, (2) pour its contents into a target receptacle, and (3) place the container at a designated location. This task requires maintaining precise control across multiple object poses and transitions:
>
> | Task            | Iter. | O | E | P | Configs | Traj. | SR in TOEP (%) |
> |-----------------|-------|---|---|---|---------|-------|----------------|
> | Pick-Pour-Place | i=1   | 1 | 1 | 1 | 1       | 20    | 30.4           |
> |                 | i=2   | 2 | 3 | 3 | 18      | 100   | 65.7           |
>
> The improvement in success rate confirms that our framework can also accommodate longer-horizon, multi-stage manipulation tasks.
>
> We view extending DexFlyWheel to longer-horizon, multi-stage manipulation tasks as an important and practical next step. In future work, we plan to integrate hierarchical policy learning into our residual framework, enabling temporal abstraction and skill composition.
>
> ---
> Hope the above clarifications and responses address your concerns. Thank you again for your time and effort!

---

> > ### Author Response · Authors · 2025-08-07
> >
> > Dear Reviewer 5Q2M,
> >
> > I’m writing to kindly follow up on our previous discussion regarding the review of our paper. We truly appreciate the time and effort you’ve dedicated to the review process.
> >
> > We have carefully addressed the concerns you raised in our rebuttal with detailed responses. If you are still considering any possible adjustments or have any further questions for us, we would be more than happy to provide any clarification or additional information needed.
> >
> > We fully understand how busy this period can be, and we greatly appreciate your continued attention. Based on your constructive feedback, we have added more experimental results to enhance the robustness of our framework evaluation. We hope our revisions and clarifications can be helpful for your final assessment.
> >
> > Thank you again for your time and thoughtful feedback.
> > \
> > \
> > Best regards, \
> > The Authors

---

> > > ### Comment · Reviewer_5Q2M · 2025-08-08
> > >
> > > Thank the authors for the rebuttal. Overall, my concern about the baselines and DexFlywheel not having the single arm results, articulated objects, and long-horizon manipulation is resolved. Overall, it appears that DexFlywheel shows its diverse applicability. I would raise my score.

---

> ### Author Response · Authors · 2025-08-09
>
> Dear Reviewer 5Q2M,
>
> Thank you very much for your thoughtful consideration and positive feedback. We are glad to hear that our additional experiments and clarifications have addressed your concerns regarding baselines, single-arm results, articulated objects, and long-horizon manipulation.
>
> We truly appreciate your willingness to raise the score and your recognition of DexFlywheel’s diverse applicability. Your constructive feedback has been invaluable in improving our work.
>
> Thank you again for your time and support.
>
>
> Best regards,
> The Authors

---

### Official Review · Reviewer_KEGx · 2025-07-03

**Clarity:** 4
**Significance:** 3
**Originality:** 3
**Rating:** 5
**Confidence:** 4

**Summary:**

This paper addresses the problem of how to generate data at-scale with limited human teleoperation time. The key insight of this paper is that manipulating different objects induces minor changes in trajectories. With this, the authors present DexFlyWheel, which is a continual learning and data collection pipeline to generate diverse and high-quality dexterous manipulation data from few actual teleoperation data. To evaluate DexFlyWheel, the authors compared it against DexMimicGen, another synthetic data generation method, and showed extensive improvement for four tasks.

**Questions:**

*  How much of this method is on manipulating objects that can be well modeled by simulator? The approach relies heavily on a simulator's ability to accurately capture the motions of the object upon interaction. The current experiments are on relatively easy-to-model objects. I may decrease the score if there is no or an uncompelling written justification. I may raise my score for a good written justification, I will raise my score for a good experimental results.

* How widely applicable is this insight of "manipulating different objects only induces minor changes in the manipulation" break, for example, lifting geometrically similar but much heavier object? How does the approach respond to this? The current experiments are on relatively easy-to-model objects. I may decrease the score if there is no or an uncompelling written justification. I may raise my score for a good written justification, I will raise my score for a good experimental results.

* Please fix typo: "desgin", numbering in baselines, there are two (4)s, ".(detailed analysis in Section 5.4.)"

**Ethical Concerns:**

["NO or VERY MINOR ethics concerns only"]

**Final Justification:**

Thank you to the nice rebuttal by the authors. My questions and issues have been resolved, and I appreciated the additional experiments taken to stress test the approach.
I have raised the sub-scores accordingly.

**Limitations:**

yes

**Quality:**

4

**Strengths And Weaknesses:**

Positives
* Very impressive task results across multiple tasks, compared to baselines (DexMimicGen)
* Experimental setup is sound, covering expected baselines
* Approach is sound and intuitive (leveraging a continual learning framework while learning a residual policy to handle the slight differences seems to be a good way to decompose the problem)

Areas of improvement
* Comment and experiments showing if this is limited to manipulating things that can be well modeled by simulator? The approach relies heavily on a simulator's ability to accurately capture the motions of the object upon interaction. The current experiments are on relatively easy-to-model objects.
* Comment and experiments on where this insight of "manipulating different objects only induces minor changes in the manipulation" break, for example, lifting geometrically similar but much heavier object.
* Typo: "desgin", numbering in baselines, there are two (4)s, ".(detailed analysis in Section 5.4.)"

---

> ### Author Rebuttal · Authors · 2025-07-31
>
> # **Response to Reviewer KEGx**
> We are so grateful for such a detailed review and positive evaluation. It is encouraging to receive recognition for our core ideas, impressive performance and clear writing! We sincerely appreciate your efforts to help us improve the quality of our work. Below, we address the concerns you raised.
>
> ---
>
> ## **Q1. How much of this method is on manipulating objects that can be well modeled by simulator? The approach relies heavily on a simulator's ability to accurately capture the motions of the object upon interaction. The current experiments are on relatively easy-to-model objects.**
>
> ### **A1: Justification for Simulation Dependence**
>
> Thank you for raising this important point. Like all simulation-based data generation methods [1-3], our approach relies on the simulator’s ability to model object interactions. This is a common and necessary trade-off that enables scalable and human-effort-less data collection compared to real-world alternatives. Thanks to recent advances [4-6], modern simulators now support complex physics and contact dynamics, and have been widely used in impactful large-scale data generation works [1-3], contributing significantly to the community.
>
> We acknowledge that simulation still has limitations in modeling fine-grained physical interactions. However, this gap is narrowing rapidly with the development of high-fidelity physics engines like physX. We have added a note in the revised manuscript’s Limitations section to explicitly acknowledge this.
>
> Importantly, we also include results on more complex objects and demonstrate transferability to real hardware when the simulated dynamics are imperfect—highlighting the robustness of our method in such cases.\
> **1. Experiment: Articulated Object Manipulation Task**
> - **Setup:** Task involves opening a 1-DOF articulated door.
>
> | Task         | Iteration | O | E | P | Configs | Traj. | SR in TOEP (%) |
> |--------------|-----------|---|---|---|---------|-------|----------------|
> | Open a Door  | i=1       | 1 | 1 | 1 | 1       | 20    | 34.2           |
> |              | i=2       | 2 | 5 | 5 | 50      | 100   | 70.6           |
>
> This demonstrates that DexFlyWheel effectively handles more complex objects. \
>  **2. Experiment: Robustness to Mass Variation (Lift Task)**
> - **Setup:** Trained on the simulated "Original" box (density = 1.0). Evaluated on both unseen simulated and real-world "Heavy" boxes (density = 10.0). The geometry and appearance are identical.
>
> | Iteration           | Sim (Original) | Sim (Heavy) | Real (Original) | Real (Heavy) |
> |---------------------|----------------|-------------|-----------------|--------------|
> | Iter 1     | 86.7           | 36.7        | 70.2            | 10.3         |
> | Iter 3 | 93.3      | 80.5        | 73.8            | 70.7         |
>
> - **Key Findings:**
>   1. The residual RL module adapts well to differences in dynamics, improving generalization to heavy objects (Iteration 3).
>   2. Real-world transfer succeeds despite training-simulation mismatch, due to diverse data generated across iterations. This shows that DexFlyWheel can handle a certain degree of inaccuracies in dynamics modeling.
>
> DexFlyWheel mitigates simulation inaccuracies by generating diverse data through residual RL and augmentation. Residual RL exposes the policy to objects with varying physical properties (e.g., mass, friction), while augmentation diversifies environmental appearance and object poses, encouraging generalization beyond simulator specifics.
>
> **Revisions**
> - We have added the results of the articulated object manipulation task in Appendix A.9: Additional Tasks Performance.
> - We have included the evaluation of simulation-to-real dynamics consistency in Appendix A.10: Robustness to Simulation Inaccuracies in Object Dynamics.
>
>
> We appreciate the opportunity to clarify our method’s reliance on simulation fidelity and mitigation strategies. In future work, we plan to incorporate advances in sim-to-real transfer to further enhance robustness, and extend DexFlyWheel to more complex domains such as fluids and deformable objects [7].
>
> ## **Q2. How widely applicable is this insight of "manipulating different objects only induces minor changes in the manipulation"?**
>
> ### **A2. Experiments and Justification for the Minor Adjustment Hypothesis**
>
> Thank you for your insightful question! This core observation directly corresponds to the key insight behind our iterative, curriculum-based learning approach.
>
> This observation holds robustly when object variations fall within plausible ranges, while in extreme cases (e.g., very large density or shape differences), trajectory deviations become larger. To handle this, our method employs a curriculum mechanism that progressively introduces object diversity across iterations, enabling effective policy adaptation. As described in Sec. 5.1 and the caption of Fig. 3(b): *"Object diversity is expanded across iterations, from a single object (i=1), to geometrically similar objects (i=2), and finally to objects with diverse geometries and physical properties (i=3)."*
>
> To validate our hypothesis that object variations induce only minor trajectory adjustments, we conducted controlled experiments across two axes: (1) object mass (lift task), and (2) object shape (grasp task). We quantified trajectory deviation via JointDiff, and report success rates (SR) across curriculum iterations (i=1,2,3) alongside Residual Norm Ratio (RNR),  which measures the relative magnitude of residual corrections compared to base policy actions.
>
> **Experiment: Trajectory Difference under Varying Mass and Shape**
>
> **Table 1.1: Trajectory Difference under Varying Mass (Lift Task)**
>
> | Density (Fixed Type) | JointDiff (rad) |
> |----------------------|-----------------|
> | 0.1 (Very Light)     | 0.23        |
> | 1.0 (Original)       | –               |
> | 10.0 (Heavy)         | 0.74        |
> | 50.0 (V-Heavy)       | 1.06        |
>
>  **Table 1.2: Trajectory Difference under Varying Shape (Grasp Task)**
>
> | Object         | JointDiff (rad) |
> |----------------|-----------------|
> | Tennis Ball    | –               |
> | Bowling Ball   | 0.75           |
> | Coke Can       | 0.59           |
> | Eggplant       | 0.95           |
> | Water Bottle   | 1.18           |
>
> **Table 2: Success Rates of DexFlyWheel Policy Across Iterations and Residual Norm Ratio**
> | Condition                     | SR i=1 (%) | SR i=2 (%) | SR i=3 (%)     |
> | ----------------------------- | ---------- | ---------- | -------------- |
> | Density 1.0 (Original)        | 86.7       | 92.0       | 93.3           |
> | Density 10.0 (Heavy)          | 36.7       | 45.0       | 80.5           |
> | Density 50.0 (V-Heavy)        | 0.0        | 12.0       | 56.0           |
> | Tennis Ball (Default)         | 94.0       | 93.7       | 94.2           |
> | Bowling Ball                  | 43.1       | 78.9       | 90.0           |
> | Coke Can                      | 38.0       | 73.4       | 80.2           |
> | Eggplant                      | 20.6       | 30.8       | 70.9           |
> | Cube                          | 15.9       | 20.0       | 85.6           |
> | Residual Norm Ratio (RNR)|     -      | 14.3 ± 2.6 |16.5 ± 2.1 |
>
> RNR is computed as:
>
> $$
> \mathrm{RNR} = \frac{1}{T} \sum_{t=1}^T \frac{ \| r_t \| }{ \| b_t \| + \epsilon }
> $$
>
> where \$r_t$ is the residual action at time step $t$, $b_t$ is the base action, and $\epsilon$ is a small constant to prevent division by zero.
>
>  **Key Findings:**
> - Trajectory deviations remain modest for reasonable object variations but increase with extreme differences.
> - Success rates improve significantly over curriculum iterations, demonstrating effective policy adaptation.
> - The relatively low Residual Norm Ratio indicates that only subtle residual corrections are necessary.
> - Our curriculum design enables robust generalization across both shape and physical property variations.
>
> We hope our controlled experiments and detailed comparisons have addressed your concerns. Thank you again for your effort and for raising such insightful questions.
>
> ## **Q3. Please fix typo.**
> > **A3**: According to your valuable suggestions, we have carefully checked and corrected the errors:
> > - **Typo**: We have changed "desgin" to "design" (Sec. 5, Line 199).
> > - **Duplicate numbering**: We have renumbered the baselines correctly in Sec. 5.1 (Lines 246–249), where the duplicate "(4)" has been fixed.
>
>
> **References:** \
> [1] Mandlekar, A., et al. "MimicGen: A Data Generation System for Scalable Robot Learning using Human Demonstrations", CoRL 2023.  \
> [2] Wang, R., et al. "DexGraspNet: A Large-Scale Robotic Dexterous Grasp Dataset for General Objects Based on Simulation", ICRA 2023.  \
> [3] Jiang, Z., et al. "DexMimicGen: Automated Data Generation for Bimanual Dexterous Manipulation via Imitation Learning", ICRA 2025.  \
> [4] Li, C., et al. "BEHAVIOR-1K: A Human-Centered, Embodied AI Benchmark with 1,000 Everyday Activities and Realistic Simulation", arXiv preprint arXiv:2403.09227, 2024.  \
> [5] Tao, S., Xiang, F., Shukla, A., et al. "ManiSkill3: GPU Parallelized Robotics Simulation and Rendering for Generalizable Embodied AI", arXiv preprint arXiv:2410.00123, revised May 2025.  \
> [6] Todorov, E., Erez, T., and Tassa, Y. "MuJoCo: A Physics Engine for Model-Based Control", IEEE/RSJ IROS, 2012, pp. 5026–5033. doi: 10.1109/IROS.2012.6386109.  \
> [7] Wang, Y., et al. “DexGarmentLab: Dexterous Garment Manipulation Environment with Generalizable Policy”, arXiv:2505.11032, 2025.

---

> > ### Comment · Reviewer_KEGx · 2025-08-05
> >
> > Thank you to the authors for the nice rebuttal.
> >
> > Q1. How much of this method is on manipulating objects that can be well modeled by simulator? The approach relies heavily on a simulator's ability to accurately capture the motions of the object upon interaction. The current experiments are on relatively easy-to-model objects.
> >
> > Thank you to the nice response and experiments on this. Please include it in the paper and also your statement that the simulation gap is decreasing.
> >
> > Q2. How widely applicable is this insight of "manipulating different objects only induces minor changes in the manipulation"?
> > Thank you for the nice experiment conducted here. I especially appreciated your experimental setup where you gradually increase across two axes. Can you comment on what manipulation tasks / objects this assumption may fail? This will help guide related works research.

---

> ### Author Response · Authors · 2025-08-06
>
> Dear Reviewer KEGx,
>
> Thank you for your positive follow-up and for highlighting the value of including our additional experiments. We truly appreciate your constructive suggestions and recognition of our efforts.
>
> **Q1.** \
> Due to rebuttal character limits, we were not able to fully elaborate in our previous response. We now provide a detailed description of the planned manuscript revisions addressing your comments:
>
> **(1) Add experimental results.**
> We have added the articulated object manipulation task **(Appendix A.9: Additional Tasks Performance)** and the evaluation of the sim-to-real dynamics gap **(Appendix A.10: Robustness to Simulation Inaccuracies in Object Dynamics)**. These results demonstrate that DexFlyWheel can effectively handle more complex articulated objects and remains robust when there are sim-to-real gap between simulated and real-world dynamics.
>
> **(2) Highlighting simulator capabilities.** To emphasize that our simulation environment supports the required level of fidelity and diversity for high-quality data generation, we added the following sentence in Section 5.1 (lines 218–220):
> > “To ensure the generation of high-quality data, we employed OmniGibson [49] as our simulation platform, which is specifically engineered to support high-quality rendering, high-precision object dynamics modeling, and a rich asset library, making it suitable for our goal.”
>
> **(3) Add statement about the narrowing sim-to-real gap.** We updated Section 6 (Limitations and Future Work) in the revised manuscript to explicitly note the narrowing sim-to-real gap:
> > "There are several limitations of our work. First, our reinforcement learning process still requires manually designed rewards, which can be time-consuming and may introduce biases. Future research could explore leveraging large language models to automate reward generation for diverse manipulation tasks. Second, our policies and simulations currently lack tactile feedback due to the immaturity of tactile sensing and simulation technologies. We plan to explore the potential of sensor-based tactile signals for contact-rich tasks in future work. **Third, our current framework relies on simulator fidelity for high-quality data generation; however, advances in simulation and sim-to-real transfer techniques are narrowing the sim-to-real gap, further enabling real-world applicability of simulation-generated data.**"
>
> **Q2.** \
> Thank you for your insightful comment. Your question about when this assumption might not hold is very helpful for guiding future research. We are glad to provide a more detailed clarification below to address this concern:
>
> > " **Where the assumption holds:** Tasks involving manipulation of different objects that share similar interaction modes and functional goals. Our experiments validate this assumption across a broad range of common dexterous manipulation tasks (e.g., grasping, pouring, lifting, handover, and articulated object manipulation), demonstrating robust performance under reasonable variations in object categories, geometries, and object properties (e.g., density 1.0–50.0). \
> **Where the assumption may fail:** Tasks involving manipulation of different objects that require distinct, object-specific strategies. In particular, for tasks demanding specialized manipulation strategies (e.g., cloth folding), changes in the manipulated objects often lead to substantial trajectory deviations because the underlying control strategy needs to change fundamentally rather than simply adjust existing trajectories. Extending our approach to tasks involving deformable object manipulation represents a promising direction for future research."
>
> **We will incorporate this clarification, along with supporting experimental details (as in our A2 response), into Appendix A.11: Applicability of the Minor Adjustment Observation of the revised manuscript.**
>
> ---
>
> Thank you once again for your insightful feedback and for guiding us in improving the paper! Given your recognition of our strong task results and sound ideas, along with the substantial improvements made based on your constructive feedback, we believe the strengthened contributions, together with the expanded evaluations and deeper analysis, now present a more robust and compelling manuscript. We would be deeply grateful if your final evaluation could take these resolutions into consideration.

---

### Official Review · Reviewer_ewWr · 2025-07-04

**Clarity:** 3
**Significance:** 3
**Originality:** 3
**Rating:** 5
**Confidence:** 4

**Summary:**

This paper presents DexFlyWheel, a scalable framework for generating diverse and high-quality dexterous manipulation data in simulation. It starts from a single human demonstration, and then iteratively expands data diversity using a closed-loop pipeline that combines imitation learning, residual reinforcement learning, trajectory rollouts, and data augmentation. The key insight is to use imitation learning to preserve human-like behaviors and using RL and domain randomization to improve generalization. The system generates over 2,000 high-quality demonstrations per task and significantly improves policy success rates in both simulation and real-world evaluations.

**Questions:**

How does the performance of DexFlyWheel scale with the number of iterations? Is there a diminishing return beyond iteration 3?

What was the wall-clock time and compute cost required to complete the three-iteration flywheel for one task?

Is there a reason the real-world deployment only includes bimanual tasks? Would the single-arm grasp/pour tasks work as well?

**Ethical Concerns:**

["NO or VERY MINOR ethics concerns only"]

**Final Justification:**

The rebuttal effectively addresses prior concerns with substantial new results: extended iteration scaling, compute/time analysis, single-arm real-world validation, qualitative failure cases, and cross-view/embodiment generalization. These additions improve both clarity and empirical support. The approach remains well-motivated, technically sound, and impactful for scalable dexterous manipulation data generation. I maintain my positive assessment and recommend acceptance.

**Limitations:**

As acknowledged by the authors, the main limitations are (1) reliance on hand-crafted reward functions for residual RL, and (2) lack of tactile sensing for contact-rich tasks. Additionally, DexFlyWheel has not yet demonstrated generalization across robot morphologies or sensor setups.

**Paper Formatting Concerns:**

No concerns.

**Quality:**

3

**Strengths And Weaknesses:**

(+) The paper presents a compelling and modular pipeline that effectively combines imitation learning, residual RL, and augmentation. The architecture (Figure 2) is clear and reproducible.

(+) The system achieves up to 81.9% success rate on challenging generalization test sets (including novel objects, environments, and spatial setups), and outperforms strong baselines (e.g., DexMimicGen) by 28.7% on average.

(+) The authors validate the pipeline with real-world bimanual robot experiments, achieving high success rates on lift and handover tasks, which is a strong testament to the practicality of the proposed approach.

(+) The contributions of the residual RL module and augmentation module are carefully disentangled through well-structured ablation studies (Table 1, Figure 5), and failure cases are systematically explored (Table 3).

(-) The framework still requires manually defined task-specific reward functions for residual RL. This contradicts the goal of scaling with minimal human input and could limit generalization to new task families.

(-) The system only relies on vision, object state, and proprioception. While well-justified for scalability, this might limit performance on fine-grained, contact-rich tasks that require force or tactile cues.

(-) While Table 3 quantifies collection success rates, the paper lacks qualitative analysis or visualizations of typical failure modes during data generation or real-world execution.

(-) It would be interesting to assess whether policies trained in one setting (e.g., robot hand or camera view) generalize to another, given the framework's goal of large-scale data generation.

---

> ### Author Rebuttal · Authors · 2025-07-31
>
> # **Response to Reviewer ewWr**
> Thank you so much for your thoughtful and constructive feedback. We are delighted that you found our method compelling, the experimental evaluation thorough, and the presentation clear! Below, we address the concerns you raised.
>
> ---
>
> ## **Q1. Performance scaling and diminishing returns**
>
> ### **A1:**
>
> The submission manuscript already consider this (Section 5.2, Table 1; the last row shows an average improvement rate of 396.4% across four tasks as the iteration number increases from 1 to 3). To provide a more detailed understanding of the performance trend, we conducted additional experiments and report the following:
>
> **(1) Performance scaling across iterations**
>
> We introduce the *Marginal Improvement (MI)* metric based on the Success Rate of $\pi_{\text{combined}}$ on $T_{\text{OEP}}$. \
> MIᵢ = SRᵢ − SRᵢ₋₁.
>
> | Iteration | SR (%) | MI (%)       |
> |-----------|--------|--------------|
> | i = 1     | 16.5   | –            |
> | i = 2     | 43.9   | +27.4        |
> | i = 3     | 81.9   | +38.0        |
>
> We observe consistent performance gains across iterations, with particularly strong improvement in iteration 3.\
> **(2) Evaluation beyond iteration 3 (up to i = 5)**
>
> | Iteration | Grasp SR (%)         | Lift SR (%)          |
> |-----------|----------------------|----------------------|
> | i = 1     | 15.0 ± 2.1           | 13.9 ± 2.8           |
> | i = 2     | 58.0 ± 4.8 (+43.0%)  | 44.4 ± 4.6 (+30.5%)  |
> | i = 3     | 90.0 ± 3.2 (+32.0%)  | 79.4 ± 7.9 (+35.0%)  |
> | i = 4     | 92.5 ± 2.8 (+2.5%)   | 82.1 ± 6.5 (+2.7%)   |
> | i = 5     | 93.2 ± 2.5 (+0.7%)   | 83.5 ± 5.8 (+1.4%)   |
>
> These results suggest that iteration *i = 3* is a practical balance between gain and cost. And the continued performance increase in later iterations highlights the potential of our framework to keep enhancing data diversity. We have added the extended results to Appendix A.5: Performance Scaling and Extended Iteration Analysis.
>
> ---
>
> ## **Q2. Wall-clock Time and Compute Cost**
>
> ### **A2:**
>
> We report the average wall-clock time and compute cost of completing the full three-iteration DexFlyWheel process across four tasks.
>
> **(1) Training Compute Cost:**
>
> - IL module: 8× NVIDIA A100 GPUs
> - Residual RL: 1× NVIDIA RTX 4090 GPU
>
> **(2) Wall-clock Training Time:**
>
> | **Policy Training (Iter.)** | **Wall-clock Time** |
> |-----------------------------|----------------------|
> | Base policy (IL)            | 5h 40m               |
> | Residual policy (RL)        | 6h 30m               |
> | **Total (3 iterations)**    | ~30 hours            |
>
> *Note: the first iteration uses only IL.*
>
> **(3) Data Collection Time (500 trajectories):**
>
> | Method              | Time        |
> |---------------------|-------------|
> | Human Teleoperation | 12h 30m     |
> | DexMimicGen         | 4h 24m      |
> | DexFlyWheel (Ours)  | **2h 24m**  |
>
> These results demonstrate the practicality and scalability of our method in generating large-scale, high-quality dexterous manipulation data with reduced human effort and modest compute requirements. We include these details in Appendix A.6: Training and Data Collection Wall-clock Time.
>
> ---
>
> ## **Q3. Is there a reason the real-world deployment only includes bi-manual tasks? Would the single-arm grasp/pour tasks work as well?**
>
> ### **A3:**
>
> We initially chose to showcase bi-manual tasks, as they better highlight the capabilities of DexFlyWheel in handling complex dexterous scenarios. Following your suggestion, we added real-world experiments for single-arm tasks and the baseline method following the same experimental setting in Section 5.5. All results have been updated in Section 5.5 Table 4.
>
> **Revised Section 5.5:**
> **Table 4. Real-world performance comparison across four tasks**
>
> | Method         | Single-arm Grasp (%) | Single-arm Pour (%) | Dual-arm Lift (%) | Dual-arm Handover (%) | Average (%) |
> |----------------|----------------------|----------------------|--------------------|------------------------|-------------|
> | DexmimicGen    | 70.5 ± 2.3           | 35.6 ± 3.1           | 59.1 ± 2.8         | 10.2 ± 1.9             | 43.85 ± 2.5 |
> | Ours           | 79.8 ± 1.7           | 76.4 ± 2.4           | 78.3 ± 1.9         | 63.3 ± 2.7             | 74.45 ± 2.2 |
>
> As shown in Table 4, our method achieves robust performance in both single-arm and dual-arm tasks, especially on more complex dual-arm tasks (e.g., +53.1% on handover vs. baseline). This demonstrates the robustness and generalization ability of DexFlyWheel in real-world transfer.
>
> ---
>
> ## **W1. The framework still requires manually defined task-specific reward functions for residual RL.**
>
> ### **A4:**
>
> As shown in the Lift Task experiment below, while we use standard reward functions for efficiency, DexFlyWheel is compatible with sparse and LLM-driven reward setups.
>
> | Method                    | SR (%)       | Training Steps |
> |---------------------------|--------------|----------------|
> | Human-design rewards [1]  | **98.0 ± 2.1**   | **50w**            |
> | LLM-driven rewards [3]    | 80.9 ± 5.7   | 100w           |
> | Sparse reward             | 84.5 ± 3.6   | 200w           |
>
> For many common dexterous manipulation tasks, well-established reward functions already exist and have been widely adopted [1]. Applying these mature reward formulations in DexFlyWheel is more efficient than sparse reward or LLM-driven rewards, due to its faster and more stable policy training **(2~4 times fewer training steps)**. There is our trade-off in minimal human input and efficient flywheel iterations. In the furture, we plan to explore how LLM-based reward generation methods [2–3] can be efficiently integrated into our framework.
>
> ---
>
> ## **W2. The system only relies on vision, object state, and proprioception. While well-justified for scalability, this might limit performance on fine-grained, contact-rich tasks that require force or tactile cues.**
>
> ### **A5:**
>
> Thank you for this question. Our framework is readily extendable to support additional modalities such as force or tactile signals.
> Specifically, we have already explored obtaining contact force signals from the simulator’s physics engine to emulate tactile interactions. However, these signals often lack realism and do not align well with the noise profile of physical tactile sensors, limiting their direct utility. Given recent advances in visual-tactile policy learning [4], we plan to investigate the integration of realistic tactile signals, potentially via calibrated sensor models or sim-to-real sensor alignment, to further strengthen our framework’s applicability to fine-grained, contact-intensive tasks.
>
> ---
>
> ## **W3. Lack qualitative analysis or visualizations of failure cases.**
> We have already provided visualizations of these failure modes in **our supplementary video (timestamp 1:45–2:04) and discussed representative failure cases in Section 5.4 (lines 303–305).**  Following your valuable suggestion, we have added Appendix A.7 to include snapshots and qualitative discussion of common failure scenarios in the revised manuscript.
>
> ### **A6:**
> **Appendix A.7 Failure Cases Analysis** \
> **(1) Failure Cases in Baseline:**
>
> 1. Generalization to Novel Geometries: Replay-based method often fails on objects with different geometries (e.g., from a ball to bottles) due to the lack of exploration mechanism.
> 2. Dynamic Contact Sensitivity: In high-dynamics tasks (e.g., mid-air handovers), even small discrepancies in object rotation, timing, or hand trajectory can lead to unstable grasp or failed transfer.
>
> **(2) Failure Cases in Our Method:**
>
> 1. IK Instabilities: In some dual-arm tasks, the solver occasionally hits singularities (e.g., elbow lockout), causing unstable or failed motions.
> 2. Out-of-Workspace: Objects at edges result in unfeasible plans.
>
> Due to rebuttal limits, visualizations will be included in the revised manuscript.
>
> ---
>
> ## **W4. Do policies trained under one embodiment or camera view generalize to others?**
>
> ### **A7:**
>
> Our dataset includes two robot hands (Psibot and Inspire) and multiple camera views (front, top, wrist). To assess generalization, we conducted two evaluations:
>
> (a) Cross-Viewpoint (Train: Front, Eval: Top):
>
> | Task  | Front View | Top View       |
> |-------|------------|----------------|
> | Grasp | 90.0 ± 3.2 | 32.3 ± 5.1     |
> | Pour  | 85.8 ± 3.5 | 45.3 ± 7.6     |
>
> (b) Cross-Embodiment (Train: Inspire, Eval: Psibot):
>
> | Task  | Inspire      | Psibot        |
> |-------|------------|----------------|
> | Grasp | 90.0 ± 3.2 | 40.6 ± 4.8     |
> | Pour  | 85.8 ± 3.5 | 38.2 ± 6.0     |
>
> Despite any optimization, policies show partial transfer, indicating the framework’s robustness. We agree this is a valuable direction and plan to explore joint cross-view and cross-embodiment training. We've included these results in revised manuscript as Appendix A.8: Evaluation of Cross-Viewpoint and Cross-Embodiment Generalization.
>
> ---
>
> **References:**
>
> [1] Y. Chen et al., "Bi-DexHands: Towards human-level bimanual dexterous manipulation," IEEE Transactions on Pattern Analysis and Machine Intelligence, vol. 46, no. 5, pp. 1-15, May 2024.\
> [2] Y. J. Ma et al., "Eureka: Human-level reward design via coding large language models," in Proc. Int. Conf. Learn. Representations (ICLR), Jan. 2024.\
> [3] T. Xie et al., "Text2Reward: Reward shaping with language models for reinforcement learning," in Proc. Int. Conf. Learn. Representations (ICLR), Jan. 2024.\
> [4] H. Xue, "Reactive Diffusion Policy: Slow-Fast Visual-Tactile Policy Learning for Contact-Rich Manipulation," in *Proc. Robotics: Science and Systems (RSS)*, 2025.

---

> ### Author Response · Authors · 2025-08-07
>
> Dear Reviewer ewWr,
>
> I’m writing to kindly follow up on our previous discussion regarding the review of our paper. We sincerely appreciate the thoughtful and constructive feedback you provided during the review process.
>
> We have carefully addressed your concerns in our rebuttal, including:
>
> - Conducting additional experiments to analyze performance scaling beyond iteration 3, demonstrating consistent improvements and validating practical trade-offs.
> - Reporting detailed wall-clock time and compute costs to demonstrate the practicality and scalability of our method.
> - Adding real-world single-arm task experiments following your suggestion, showing robust performance in both single-arm and dual-arm scenarios.
> - Discussing the compatibility with different reward functions and future plans for integrating LLM-based rewards.
> - Providing qualitative failure case analysis.
> - Evaluating cross-viewpoint and cross-embodiment generalization.
>
> If you have any further questions or require clarifications, we would be happy to assist.
>
> We fully understand how busy this period can be and greatly appreciate your time and attention. We would be deeply grateful if you could take these revisions and clarifications into consideration during your final evaluation.
>
> Thank you again for your thoughtful feedback.
>
> Best regards,
> The Authors

---

### Comment · Area_Chair_B61t · 2025-08-05

Dear Reviewers,

The authors have responded to the original reviews. Could you read the rebuttal and share your thoughts? Does it address your original concerns? Are there any remaining questions for the authors?

Best,
AC

---

### Note · Authors · 2025-08-16

Dear AC and Reviewers,

We sincerely thank you for your time, thoughtful feedback, and constructive discussions, which have been instrumental in strengthening our paper.

---

# 1. Strengths Highlighted by Reviewers

We are encouraged that the reviewers recognized the strengths of our work: Reviewer ewWr highlighted our pipeline as **compelling** and a **strong testament** to its **practicality**, while Reviewer KEGx noted that it is **sound and intuitive**, offering a **good way to decompose the problem** with **very impressive results**. Reviewer 5Q2M further emphasized its **high scalability** and **promising sim-to-real transfer**, and Reviewer XGgH also acknowledged the **meaningful results**.



---

# 2. Our Responses and Discussions

We carefully addressed all reviewer concerns through additional experiments and clarifications. Specifically, we added results with different reward functions, tested single-arm and baseline settings, and included articulated objects and long-horizon tasks. Reviewer XGgH noted, **“I believe that this rebuttal has sufficiently addressed these concerns”**, and further mentioned **“I increase my score”**, while Reviewer 5Q2M stated, **“my concern … is resolved. I would raise my score.”** Reviewer KEGx also responded with **“nice rebuttal.”**

Meanwhile, following Reviewer KEGx’s suggestion, we clarified the scope under which our assumptions hold and do not hold, helping guide related research.

Additionally, we reported further results to address Reviewer ewWr’s concerns on performance scaling, wall-clock time, computational costs, as well as cross-viewpoint and cross-embodiment generalization.

Finally, following Reviewer XGgH’s advice, we added continual reinforcement learning comparisons, upon which Reviewer XGgH remarked, **“A4 provides a compelling result … which removes my concerns.”**



---

# 3. Closing Statement

We believe our work provides a significant step toward building scalable and transferable solutions for dexterous manipulation data generation. The proposed framework demonstrates both compelling data generation performance across different tasks and strong potential for sim-to-real transfer.

As the rebuttal has addressed all major concerns and resolved specific issues raised by the reviewers, we sincerely thank the AC and reviewers again for their constructive engagement, and we look forward to a favorable consideration.

---

### Decision · Program_Chairs · 2025-09-17

**Decision:**

Accept (spotlight)

**Comment:**

This paper presents a scalable approach for generating diverse and high quality demonstrations in simulation for dexterous manipulation tasks. The pipeline can start with as little as a single human demonstration and through a self-improving cycle, it can gradually increase the high quality demonstrations to up to 2000. Initially, the paper received one Reject, one Borderline Accept and two Accept ratings. The reviewers were impressed with the performance of the proposed pipeline compared to previous work, its scalability and practicality. However there were some remaining questions, e.g., related to scope, comparison with prior work, reliance on manually defined task-specific reward functions and limitations imposed by the capabilities of the simulator. During the rebuttal period, the authors replied to the concerns of the reviewers, with additional experiments and clarifications. The rebuttal answers were received very well by the reviewers and eventually they increased their ratings to one Borderline Accept and three Accept ratings. Given the strong and unanimous support from four knowledgable reviewers, there is no basis to overturn reviews. This is a well-motivated approach with strong results that has the potential to be quite impactful. The AC recommends acceptance. Authors should still consider any additional comments or feedback from the reviewers while preparing their final version (e.g., improve clarity), and of course update the manuscript to include any additional promised changes, analysis, results and/or discussion they provided in their rebuttal.